

**Heat stored in the Earth system 1960-2020: Where does the energy go?**
**Authors:** Karina von Schuckmann[1], Audrey Minère[1], Flora Gues[1,2], Francisco José Cuesta
Valero[3,36], Gottfried Kirchengast[4], Susheel Adusumilli[5], Fiamma Straneo[5], Richard P. Allan[6], Paul
M. Barker[7], Hugo Beltrami[8,51], Tim Boyer[9], Lijing Cheng[10,11], John A. Church[7], Damien
Desbruyeres[12], Han Dolman[13], Catia M. Domingues[14], Almudena García-García[3,36], Donata
Giglio[15], John E. Gilson[5], Maximilian Gorfer[16,4], Leopold Haimberger[17], Stefan Hendricks[18],
Shigeki Hosoda[19], Gregory C. Johnson[20], Rachel Killick[21], Brian King[14], Nicolas Kolodziejczyk[22],
Anton Korosov[23], Gerhard Krinner[24], Mikael Kuusela[25], Moritz Langer[26,27], Thomas Lavergne[28],
Isobel Lawrence[29], Yuehua Li[30], John Lyman[20], Ben Marzeion[21], Michael Mayer[17,31], Andrew H.
MacDougall[32], Trevor McDougall[7], Didier Paolo Monselesan[33], Jan Nitzbon[26,34], Inès Otosaka[35],
Jian Peng[3,36], Sarah Purkey[5], Dean Roemmich[5], Kanako Sato[19], Katsunari Sato[37], Abhishek
Savita[38], Axel Schweiger[39], Andrew Shepherd[35], Sonia I. Seneviratne[40], Leon Simons[41], Donald
A. Slater[42], Thomas Slater[35], Noah Smith[43], Andrea Steiner[4], Toshio Suga[44,19], Tanguy Szekely[45],
Wim Thiery[46], Mary-Louise Timmermans[47], Inne Vanderkelen[46], Susan E. Wjiffels[33,48], Tonghua
Wu[49], Michael Zemp[50]
Corresponding author: Karina von Schuckmann, karina.von.schuckmann@mercator-ocean.fr
Mercator Ocean International, Toulouse, France
CELAD, Toulouse, France
Department of Remote Sensing, Helmholtz Centre for Environmental Research, Leipzig, 04318, Germany
Wegener Center for Climate and Global Change and Institute of Physics, University of Graz, Graz, Austria
Scripps Institution of Oceanography, University of California San Diego, San Diego, California, USA
University of Reading, UK
University of New South Wales, Sydney, Australia
Climate & Atmospheric Sciences Institute and Department of Earth Sciences, St. Francis Xavier University,
Antigonish, B2G 2W5, Canada
NOAA's National Centers for Environmental Information, Silver Spring, Maryland, USA
Institute of Atmospheric Physics, Chinese Academy of Sciences, Beijing, China
Center for Ocean Mega-Science, Chinese Academy of Sciences, Qingdao, 266071, China
Ifremer, University of Brest, CNRS, IRD, Laboratoire d'Océanographie Physique et Spatiale, Brest, France
Netherlands Institute for Sea Research, Den Burg, Texel, Netherlands
National Oceanographic Centre, Southampton, UK
University of Colorado, Boulder, USA
Center for Climate Systems Modeling, ETH Zurich, Zurich, Switzerland
Department of Meteorology and Geophysics, University of Vienna, Vienna, Austria
Alfred Wegener Institute Helmholtz Centre for Polar and Marine Research, Germany
Japan Marine-Earth Science and Technology (JAMSTEC), Japan
NOAA, Pacific Marine Environmental Laboratory, Seattle, USA
Met Office Hadley Centre, Exeter, UK
University of Brest, CNRS, IRD, Ifremer, Laboratoire d'Océanographie Physique et Spatiale, IUEM, Brest,
France
Nansen Environmental and Remote Sensing Center, Norway
Institut des Géosciences de l'Environnement, CNRS, Université Grenoble Alpes, Grenoble, France
Carnegie Mellon University, Pittsburg, USA
Alfred Wegener Institute Helmholtz Centre for Polar and Marine Research, Permafrost Research Section,
Potsdam, Germany
Humboldt-Universität zu Berlin, Geography Department, Berlin, Germany
Norwegian Meteorological Institute, Norway
European Space Agency, ESRIN, Via Galileo Galilei, 1, 00044 Frascati RM, Italy
University of Bremen, Germany



European Centre for Medium-Range Weather Forecasts (ECMWF), Reading, UK
Climate & Environment Program, St. Francis Xavier University Antigonish, Nova Scotia, Canada B2G 2W5
CSIRO Oceans and Atmosphere, Hobart, Tasmania, Australia
Alfred Wegener Institute Helmholtz Centre for Polar and Marine Research, Paleoclimate Dynamics Section, Bremerhaven, Germany.
Centre for Polar Observation and Modelling, University of Leeds, UK
Remote Sensing Centre for Earth System Research, Leipzig University, 04103, Leipzig, Germany
Japan Meteorological Agency, Japan
GEOMAR, Kiel, Germany
Polar Science Center, Applied Physics Laboratory, University of Washington, Seattle, WA, USA
Institute for Atmospheric and Climate Science, ETH Zurich, Zurich, 8092, Switzerland
The Club of Rome, The Netherlands Association, 's-Hertogenbosch, The Netherlands
Glaciology and Oceanography, Univ. of Edinburgh, UK
Department of Mathematics, University of Exeter, Exeter, United Kingdom
Tohoku University, Japan
Ocean Scope, Brest, France
Department of Hydrology and Hydraulic Engineering, Vrije Universiteit Brussel, Brussels, 1050, Belgium
Department of Earth and Planetary Sciences, Yale University, New Haven, Connecticut, USA
Woods Hole Oceanographic Institution, Massachusetts, USA
Cryosphere Research Station on Qinghai–Xizang Plateau, State Key Laboratory of Cryospheric Science, Northwest Institute of Eco–Environment and Resources (NIEER), Chinese Academy of Sciences (CAS), Lanzhou, 730000, China
Department of Geography, University of Zurich, Switzerland
Département des sciences de la Terre et de l'atmosphère, Université du Québec à Montréal, Montréal, Québec, Canada.

**Abstract.** The Earth climate system is out of energy balance and heat has accumulated continuously over the past decades, warming the ocean, the land, the cryosphere and the atmosphere. According to the 6th Assessment Report of the Intergovernmental Panel on Climate Change, this planetary warming over multiple decades is human-driven and results in unprecedented and committed changes to the Earth system, with adverse impacts for ecosystems and human systems. The Earth heat inventory provides a measure of the Earth energy imbalance, and allows for quantifying how much heat has accumulated in the Earth system, and where the heat is stored. Here we show that $380 \pm 62$ ZJ of heat has accumulated in the Earth system from 1971 to 2020, at a rate of $0.48 \pm 0.1$ W m$^{-2}$, with $89 \pm 17$ % of this heat stored in the ocean, $6 \pm 0.1$ % on land, $4 \pm 1$% in the cryosphere and $1 \pm 0.2$ % in the atmosphere. Over the most recent decade (2006-2020), the Earth heat inventory shows increased warming at rate of $0.48 \pm 0.3$ W m$^{-2}$/decade, and the Earth climate system is out of energy balance by $0.76 \pm 0.2$ Wm$^{-2}$. The Earth heat inventory is the most fundamental global climate indicator that the scientific community and the public can use as the measure of how well the world is doing in the task of bringing anthropogenic climate change under control. We call for an implementation of the Earth heat inventory into the Paris agreement's global stocktake based on best available science. The Earth heat inventory in this study, updated from von Schuckmann et al, 2020, is underpinned by worldwide multidisciplinary collaboration and demonstrates the critical importance of concerted international efforts for climate change monitoring and community-based recommendations as coordinated by the Global Climate Observing System (GCOS). We also call for urgently needed actions for enabling continuity, archiving, rescuing and calibrating efforts to assure improved and long-term monitoring capacity of the relevant GCOS Essential Climate Variables (ECV) for the Earth heat inventory.



## Introduction

Since a recent international quantification of the Earth heat inventory (von Schuckmann et al., 2020), three main reports of the 6th assessment cycle of the Intergovernmental Panel for Climate Change (IPCC)[1] have been published. The IPCC report of Working Group III (WGIII) 'Climate Change 2022: Mitigation of Climate Change' (IPCC, 2022b) states that '*options available now in every sector that can at least halve emissions by 2030*' and that '*accelerated climate action is critical to sustainable development*'[2]. The IPCC report of Working Group II (WGII) 'Climate Change 2022: Impacts, Adaptation and Vulnerability' (IPCC, 2022a) offers solutions, while pointing out that '*every small increase in warming will result in increased risks*', and that '*it is essential to make rapid, deep cuts in greenhouse gas emissions to keep the maximum number of adaptation options open*[3]. The IPCC report of Working Group I (WGI) 'Climate Change 2022: The Physical Science Basis' (IPCC, 2021) concluded that '*recent human-induced changes in the climate are widespread, rapid, and intensifying, and unprecedented in thousands of years*', and '*that there is no going back from some changes in the climate system, from which some changes could be slowed and others could be stopped by limiting warming*'[4].

These assessment outcomes further emphasize the need to extend the Global Climate Observing System (GCOS) beyond the strict scientific observation of the climate state to also supporting policy and planning (GCOS, 2021). The GCOS was established in 1992 to aid in developing and coordinating a GCOS that supported scientific understanding of climate change. More recently it has broadened its focus to include policy development, public information and planning for adaptation and mitigation (GCOS, 2016). GCOS started assessments of the Earth's heat inventory in 2018, and the carbon and the water cycles, to identify potential gaps and inconsistencies in existing observation systems (Crisp et al., 2022; Dorigo et al., 2021; von Schuckmann et al., 2020). The first call for concerted international collaboration on the Earth's energy imbalance and the associated Earth heat inventory had been established in a perspective paper in 2016 (von Schuckmann et al., 2016), initiating a research focus activity under WCRP/CLIVAR[5]. One of the outcomes was the development of an internationally and multidisciplinary driven publication on the Earth heat inventory, now under the auspices of GCOS (von Schuckmann et al., 2020), which further continues with this study. With this second study we aim to contribute to a more frequent and regular update of the state of the Earth heat inventory as an important indicator of climate change.

The Earth heat inventory provides a quantitative measure of the heat accumulated in the Earth system, which results from the anthropogenically perturbed planetary radiation budget – i.e., a positive Earth Energy Imbalance (EEI) forced by increasing atmospheric concentrations of radiatively active greenhouse gasses from human-induced emissions (Forster et al., 2022; Hansen et al., 2011) (Fig. 1). Estimates of the Earth heat inventory can be obtained by analyzing several

---

[1]     https://www.ipcc.ch/
[2]     https://report.ipcc.ch/ar6wg3/pdf/IPCC_AR6_WGIII_PressConferenceSlides.pdf
[3]     https://report.ipcc.ch/ar6wg2/pdf/IPCC_AR6_WGII_PressConferenceSlides.pdf
[4]     https://www.ipcc.ch/report/ar6/wg1/resources/presentations-and-multimedia
[5]     https://www.clivar.org/research-foci/heat-budget





Essential Climate Variables (ECVs) of GCOS, complemented by model and reanalysis outputs to
fill the gaps, through the quantification of increases in heat content of the ocean, the land, the
atmosphere, and the heat used to melt ice (Forster et al., 2022; von Schuckmann et al., 2020). This
assessment allows for evaluating the total heat accumulated in the Earth system and where and
how much heat is stored in the different Earth system components (Fig. 1). The derivative of the
Earth heat inventory over time provides then an estimate of the global heating rate, and hence, the
absolute value of the EEI (Loeb et al., 2012; Trenberth et al., 2016). A recent quantification of the
Earth heat inventory (von Schuckmann et al., 2020) revealed a consistent long-term Earth system
heat gain over the period 1971–2018, with a total heat gain of 358±37 ZJ, which is equivalent to a
global heating rate of 0.47±0.1 W m$^{-2}$. Over the period 1971–2018, the majority of heat gain is
reported for the global ocean, with 89 % of the excess heat in the climate system stored there, and
for 2010–2018 that was 90%. 52 % of the excess heat was stored in the upper 700 m of the ocean
for both time periods, with 28 % stored in the 700–2000 m depth layer and 9 % below 2000 m
depth for 1971–2018 (30% in the 700–2000 m layer and 8% below 2000 m for 2010–2018). For
1971–2018, heat gain by the land amounts to 6 % of the total, 4 % is used for the melting of
grounded and floating ice, and 1 % goes to atmospheric warming. Those fractions are 5%, 3%, and
2% respectively for 2010–2018. The results are consistent within uncertainty ranges with the
assessment outcomes as obtained in the recent IPCC report (Forster et al., 2022).
The rate of change in the Earth heat inventory, and hence, the EEI, is the portion of the forcing
that the Earth has not yet realized as warming (Hansen et al., 2005). The Earth system responds to
an imposed radiative forcing through a number of feedbacks, which operate on various different
timescales. Earth's radiative response is complex, comprising a variety of climate feedbacks (e.g.,
water vapor feedback, cloud feedbacks, ice-albedo feedback) (Forster et al., 2022). Conceptually,
the relationships between EEI, radiative forcing and surface temperature change can be expressed
as (Gregory & Andrews, 2016):
$\Delta N_{TOA} = \Delta F_{ERF} - |\alpha_{FP}|\Delta T_S$ ,                                   (1)
where $\Delta N_{TOA}$ is the Earth's net energy imbalance at the Top Of the Atmosphere (TOA) (in W m$^{-2}$),
$\Delta F_{ERF}$ is the effective radiative forcing (W m$^{-2}$), $\Delta T_S$ is the global surface temperature anomaly
(K) relative to the equilibrium state and $\alpha_{FP}$ is the net total feedback parameter (W m$^{-2}$ K$^{-1}$), which
represents the combined effect of the various climate feedbacks. Essentially, $\alpha_{FP}$ in Eq. (1) can be
viewed as a measure of how efficient the system is at restoring radiative equilibrium for a unit
surface temperature rise. Thus, $\Delta N_{TOA}$ represents the difference between the applied radiative
forcing and Earth's radiative response through climate feedbacks associated with surface
temperature increase (e.g., Hansen et al., 2011). Observation-based estimates of $\Delta N_{TOA}$ are
therefore crucial both to our understanding of past climate change and for refining projections of
future climate change (Gregory & Andrews, 2016; Kuhlbrodt & Gregory, 2012). The long
atmospheric lifetime of carbon dioxide means that $\Delta N_{TOA}$, $\Delta F_{ERF}$ and $\Delta T_S$ will remain positive for
centuries, even with substantial reductions in greenhouse gas emissions, and lead to substantial
sea-level rise, ocean warming and ice shelf loss (Cheng et al., 2019; Forster et al., 2022; Hansen
et al., 2017; IPCC, 2021; Nauels et al., 2017). In other words, warming will continue even if
atmospheric greenhouse gas (GHG) amounts are stabilized at today's level, and the EEI defines
additional global warming that will occur without further change in forcing (Hansen et al., 2017).
The EEI is less subject to decadal variations associated with internal climate variability than global





surface temperature and therefore represents a robust measure of the rate of climate change, and
its future commitment (Cheng et al., 2017; Forster et al., 2022; Palmer & McNeall, 2014; von
Schuckmann et al., 2016).

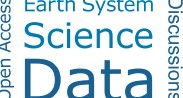




**Fig. 1:** *Schematic overview on the central role of the Earth heat inventory and its linkage to anthropogenic emissions, the Earth energy imbalance, change in the Earth system and implications for ecosystems and human systems. The Earth heat inventory plays a central role for climate change monitoring as it provides information on the absolute value of the Earth energy imbalance, the total Earth system heat gain, and how much and where heat is stored in the different Earth system components. Examples of associated global-scale changes in the Earth system as assessed in (Gulev et al., 2021) are drawn, together with major implications for the ecosystem and human systems (IPCC, 2022c). Upward arrows indicate increasing change, downward arrows indicate decreasing change, and turning arrows indicate change in both directions. The % for heat stored in the Earth system are provided over the period 2006-2020 (see section 6).*

The heat gain in the Earth system from a positive EEI results in directly and indirectly triggered changes in the climate system, with a variety of implications for the environment and human systems (Fig. 1). One of the most direct implications from a positive EEI is the rise of Global Mean Surface Temperature (GMST). The accumulation and storage of surplus anthropogenic heat leads to ocean warming and thermal expansion of the water column, which together with terrestrial ice melt leads to sea level rise (WCRP Global Sea Level Budget Group, 2018). Moreover, there are various facets of impacts from ocean warming such as on climate extremes, which are provided in more detail in a recent review (Cheng et al., 2022). The heat accumulation in the Earth system also leads to warming of the atmosphere, particularly to a temperature increase in the troposphere, leading to water vapor increase and changes in atmospheric circulation (Gulev et al., 2021).

On land, the heat accumulation leads to an increase in ground heat storage, which in turn triggers an increase in ground surface temperature that may increase soil respiration, and evaporation, and may lead to a decrease in soil water, depending on the climatic and meteorological conditions and factors such as land cover and soil characteristics (Cuesta-Valero et al., 2022; Gulev et al., 2021). Moreover, inland water heat storage increases, which in turn leads to increases in lake water temperature that may result in algal blooms and lake stratification, and typically leads to a decrease in ice cover. Heat gain in the Earth system also induces an increase in permafrost heat content, which in turn increases ground subsidence, $CH_4$ and $CO_2$ emissions, and a decrease in permafrost extent and ground ice volume. More details are synthesized in (Cuesta-Valero et al., 2022). In the cryosphere associated changes include a loss of glaciers, ice sheets and Arctic sea ice (IPCC, 2019). These human-induced changes have already impacted terrestrial, freshwater and ocean ecosystems, and have adverse impacts on human systems (Fig.1). Particularly, they have emerged for ecosystem structure, species ranges and phenology (timing of life cycles), and include adverse impacts such as for water security and food production, health and wellbeing, cities, settlements and infrastructures as assessed in detail in the recent IPCC Working Group II report (IPCC, 2022c, see their Fig. SPM.2).

In summary, the Earth heat inventory is a global climate indicator integrating fundamental aspects of the Earth system under global warming. Particularly, the global climate indicator of the Earth heat inventory

- provides the best available current estimate of the absolute value of the Earth Energy Imbalance (Cheng et al., 2017; Cheng et al., 2019; Hakuba et al., 2021; Hansen et al., 2011; Loeb et al., 2012, 2022; Trenberth et al., 2016; von Schuckmann et al., 2020),

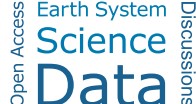

• enables an integrated view of the effective radiative climate forcing, Earth's surface
temperature response and the climate sensitivity (Forster et al., 2022; Hansen et al., 2011;
Hansen et al., 2005; Palmer & McNeall, 2014; Smith et al., 2015),
• informs about the status of global warming in the Earth system as it integrates the heat 'in
the pipeline' that will ultimately warm the deep ocean and melt ice sheets in the long term
(Hansen et al., 2011; Hansen et al., 2005; IPCC, 2021),
• reveals how much and where surplus anthropogenic heat is available for melting the
cryosphere and warming the ocean, land and atmosphere, which in turn allows for an
evaluation of associated changes in the climate system and is essential to improve seasonal-
to-decadal climate predictions and projections on century timescales to enable improved
planning for and adaptation to climate change (Hansen et al., 2011; von Schuckmann et al.,
2016, 2020),
• provides a tool for assessing the status of the GCOS, the identification of its strength and
gaps, and the development of crucial recommendations of its future evolution (GCOS,
2021; von Schuckmann et al., 2020),
• creates an opportunity for a safe climate pathway while evaluating an atmospheric $CO_2$
reduction amount to bring Earth back towards energy balance (Hansen et al., 2000; von
Schuckmann et al., 2020).
• Enables concerted international and multidisciplinary collaboration and advancements in
climate science.
Hence, regularly assessing, quantifying and evaluating the Earth heat inventory creates a unique
opportunity to support the call of action and solution pathways as assessed during the 6th
assessment cycle of the IPCC. Moreover, the Earth heat inventory allows for a regular stock taking
of the implementation of the Paris Agreement[6] while monitoring progress towards achieving the
purpose of the agreement and its long-term goals based on best available science.
Based on the quantification of the Earth heat inventory published in 2020 (von Schuckmann et al.,
2020), we will present the updated results of the Earth heat inventory over the period 1960-2020,
along with the long-term Earth's system heat gain over this period, and the partitions of where the
heat goes for the ocean, atmosphere, land and cryosphere. Section 2 provides the updates for ocean
heat content, which is based on improved evaluations and the addition of further international data
products of subsurface temperature. Updated estimates and refinements for atmospheric heat
content are discussed in Section 3. For the land component in section 4, an improved uncertainty
framework is proposed for the ground heat storage estimate, and new evaluations for inland
freshwater heat storage and thawing of permafrost have been included (Cuesta-Valero et al., 2022).
Heat available to melt the cryosphere is described in section 5. In section 6, the updated Earth heat
inventory is established and discussed based on the results of sections 2-5. In the final section,

---

6        https://unfccc.int/process-and-meetings/the-paris-agreement/the-paris-agreement



challenges and recommendations for future improved estimates are discussed for each Earth
system component, with associated recommendations for future evolutions of the GCOS.

**2. Heat stored in the ocean**

Estimating global Ocean Heat Content (OHC) directly depends on the variables of the in situ
component of the Global Ocean Observing System (GOOS), which has continued to evolve during
the past century (Abraham et al., 2013; Gould et al., 2013; Moltmann et al., 2019). Many global
OHC estimates for the historical period start from about the 1950s and 1960s, i.e., when shipboard
Nansen bottle and mechanical bathythermograph (MBT) instruments, conductivity–temperature–
depth (CTD) instruments and the expendable bathythermograph (XBT) became available
(Abraham et al., 2013; Goni et al., 2019). In the 1980s and 1990s, the GOOS (GOOS, 2019) started
to further evolve, including programs for moored arrays in the tropical ocean basins, and the
international World Ocean Circulation Experiment (WOCE) (Gould et al., 2013; King et al., 2001).
Estimates of global OHC are, however, challenged by various factors, such as limited global
coverage and data quality. The international community, especially under the auspices of the
International qualiy-controlled Ocean Database project (IQuOD[7]), works together to face these
obstacles through data and meta-data recovery and improved observational uncertainty
specification, bias correction methods, and data processing techniques (Boyer et al., 2016;
Castelao, 2020; Castelão, 2021; Cheng et al., 2018; Cowley et al., 2021; Goni et al., 2019;
Gouretski & Cheng, 2020; Leahy et al., 2018; Mieruch et al., 2021; Palmer et al., 2018; Savita et
al., 2022). Satellite altimeter measurements of sea surface height began in 1993 and are used to
complement in situ-derived OHC estimates, either for validation purposes (Cabanes et al., 2013)
or for establishing global gridded ocean temperature fields (Guinehut et al., 2012; Willis et al.,
2004). Indirect estimates of OHC from remote sensing through the global sea-level budget became
possible with satellite-derived ocean mass information in 2002 (Dieng et al., 2017; Hakuba et al.,
2021; Llovel et al., 2014; Marti et al., 2022; Meyssignac et al., 2019), and should be considered in
future establishments of the Earth heat inventory.

From the year 2000 onwards, the in situ component of the GOOS was revolutionized with the
implementation of an international program of profiling floats targeting global hydrographic
measurements of the upper 2000m depth (Riser et al., 2016; Roemmich et al., 2019) – a target
which was largely reached in 2005 for the ocean area between 60°S-60°N and fully realized in
2006 (Riser et al. 2016). The opportunity for improved OHC estimates provided by Argo is
tremendous and has led to major advancements in climate science, particularly on the discussion
of the EEI (Cheng et al., 2019; Forster et al., 2022; Hansen et al., 2011; Johnson et al., 2016; Loeb
et al., 2012, 2021; Trenberth & Fasullo, 2010). The near global coverage of the Argo network also
provides an excellent test bed for the long-term OHC reconstruction extending back well before
the Argo period (Allison et al., 2019; Cheng, Trenberth, Fasullo, Boyer, et al., 2017). Moreover,
these evaluations inform further observing system recommendations for global climate studies,
i.e., gaps in the deep ocean layers below 2000m depth, in marginal seas, in shelf areas and in the
polar regions (von Schuckmann et al., 2016; 2020). Gap implementations are underway, for
example, for the deep Argo array (Johnson et al., 2019). Different research groups have developed
gridded products of subsurface temperature fields using different processing methodologies, and

---

7     www.iquod.org



an exhaustive list can be found in (Abraham et al., 2013; Boyer et al., 2016; Savita et al., 2022;
Cheng et al., 2022; Gulev et al., 2021). Additionally, specific Argo-based products are listed on
the Argo web page (http://www.argo.ucsd.edu/, last access: 12 July 2022). Albeit the tremendous
improvement of in situ subsurface temperature measurements over time, estimates of global OHC
remain an area of active research to minimize effects from different data processing techniques of
the irregular in situ database, the choice of the climatology used in the mapping process, and data
bias corrections, which today induce discrepancies between the different estimates (Boyer et al.,
2016; Cheng et al., 2019; Good, 2017; Gouretski & Cheng, 2020; Savita et al., 2022). Ocean
reanalysis systems have also been used to deliver estimates of near-global OHC (Trenberth et al.,
2016; von Schuckmann et al., 2018), and their international assessments show increased agreement
with increasing in situ data availability for the assimilation, particularly after 2005, i.e. when Argo
had achieved nearly global scale data sampling (Palmer et al., 2017; Storto et al., 2018, 2019).
This initiative relies on the availability of regular updates of data products, their temporal
extensions and direct interactions with the different research groups. A complete view of all
subsurface ocean temperature products can be only achieved through a concerted international
effort and over time, particularly accounting for the continued development of new or improved
OHC products. In this study, we do not achieve a holistic view of all available products but present
a starting point for future international regular assessments of global OHC. A first established
international ensemble mean and standard deviation of near global OHC up to 2018 was
established in von Schuckmann et al. (2020), which has now been updated up to 2020, and further
extended with the addition of 4 new products (Fig. 3). The ensemble spread gives an indication of
the agreement among products and can be used as a proxy for uncertainty. Compared to the results
in von Schuckmann et al. (2020), the spread has increased by about 0.1 W m$^{-2}$ the recent period
2006-2020 for the 0-2000m and 700-2000m integration depth layers. Concerns about common
errors in the products remain. Accurate understanding of the uncertainties of the product is an
essential element in their use. So far, a basic assumption is that the error distribution for the
observations is Gaussian with a mean of zero, which has been approximated by an ensemble of
various products. However, a more complete understanding of any apparent trends requires
determination of systematic errors (e.g., systematic calibration errors), or the impacts of changing
observation densities, and of instrument technologies (Wong et al., 2020). These elements can
result in biases across the ensemble, or produce artificial changes in the energetics of the system
(Wunsch, 2020). The uncertainty can also be estimated in other ways including some purely
statistical methods (Cheng et al., 2019; Levitus et al., 2012; MacIntosh et al., 2017) or methods
explicitly accounting for the error sources (Gaillard et al., 2016; Lyman & Johnson, 2014; von
Schuckmann & Le Traon, 2011). Each method has its caveats; for example, the error covariances
are mostly unknown, and must be estimated a priori. For this study, adopting a straightforward
method with a "data democracy" strategy (i.e., all OHC estimates have been given equal weights)
has been chosen as a starting point, differently from the ensemble approach adopted in AR6
(Forster et al., 2022).



Open Access · Earth System Science Data · Discussions

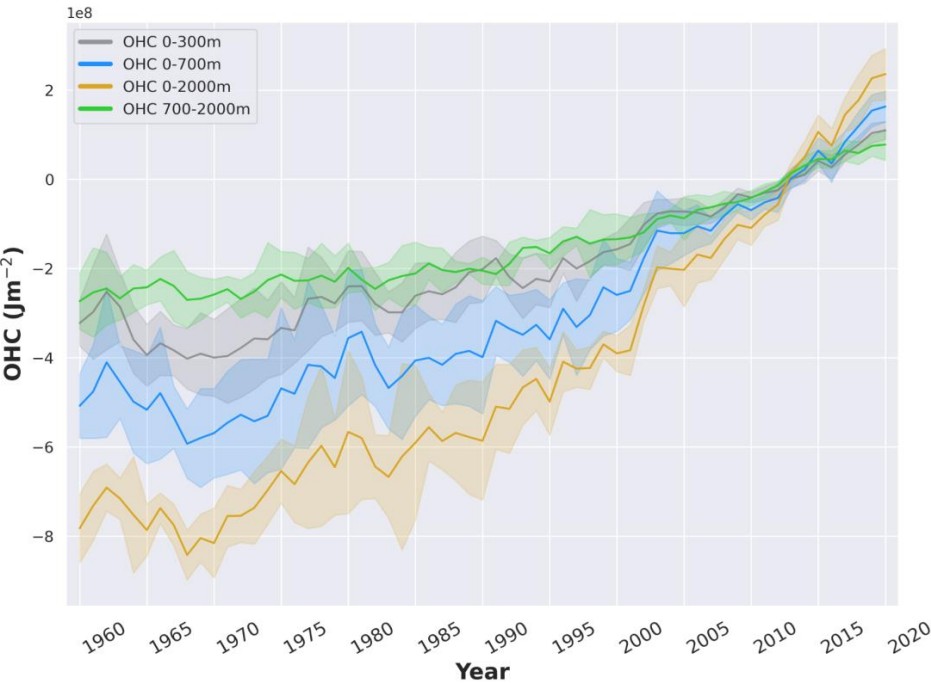

*Figure 2. Ensemble mean time series and ensemble standard deviation (2-σ, shaded) of global*
*ocean heat content (OHC) anomalies relative to the 2005–2020 climatology for the 0–300m*
*(gray), 0–700m (blue), 0–2000m (yellow) and 700–2000m depth layer (green). The ensemble mean*
*is an outcome of an international assessment initiative, and all products used are referenced in*
*the legend of Fig. 3. The trends derived from the time series are given in Table 1. Note that values*
*are given for the ocean surface area between 60°S and 60°N and are limited to the 300m*
*bathymetry of each product.*

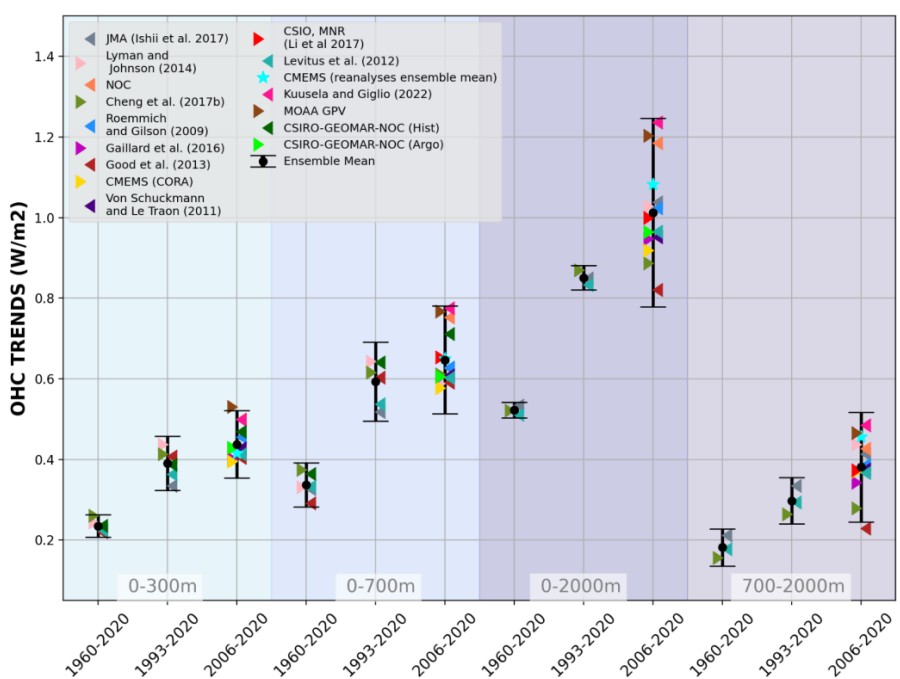

*Figure 3. Trends of global ocean heat content (OHC) as derived from different products (colors),*
*and using LOWESS (see text for more details). References are given in the figure legend, except,*
*CMEMS (CORA and ARMOR-3D, http://marine.copernicus.eu/science-learning/ocean-*
*monitoring-indicators, last access: 28 June 2022), CSIRO-GEOMAR-NOC (Argo) (Domingues et*
*al., 2008; Roemmich et al., 2015; Wijffels et al., 2016), CSIRO-GEOMAR-NOC (hist) (Church et*
*al., 2011; Domingues et al., 2008), NOC (National Oceanographic Institution) (Desbruyères et*
*al., 2017) and the Argo dataset MOAA GPV (Hosoda et al., 2008). The ensemble mean and*
*standard deviation (2 σ) are indicated in black. The shaded areas show trends from different depth*
*layer integrations, i.e., 0–300m (light turquoise), 0–700m (light blue), 0–2000m (purple) and 700–*
*2000m (light purple). For each integration depth layer, trends are evaluated over the three study*
*periods, i.e., historical (1960–2020), altimeter era (1993–2020) and golden Argo era (2006–*
*2020). See text for more details on the international assessment criteria. Note that values are given*
*for the ocean surface area (see text for more details). References as indicated in the legend include*
*(Cheng, Trenberth, Fasullo, Boyer, et al., 2017; Gaillard et al., 2016; Good et al., 2013; Ishii et*
*al., 2017; Kuusela & Giglio, 2022; Levitus et al., 2012; Li et al., 2017; Lyman & Johnson, 2014;*
*Roemmich & Gilson, 2009; von Schuckmann & Le Traon, 2011).*
The continuity of this activity will help to further expand international collaboration and to unravel
uncertainties due to the community's collective efforts on data quality as well as on detecting and
reducing processing errors (e.g., IQuOD). It also provides up-to-date scientific knowledge of ocean
warming. Products used for this assessment are referenced in the caption of Fig. 3. Estimates of
OHC have been provided by the different research groups under largely homogeneous criteria. All



estimates use a coherent ocean volume limited by the 300m isobath of each product and are limited
to 60°S–60°N since most observational products exclude high latitude ocean areas because of the
low observational coverage, and only annual averages have been used. The ocean areas within
60°S–60°N includes 91% of the global ocean surface area, and limiting to the 300m isobath
neglects the contributions from coastal and shallow waters, so the resultant OHC trends will be
underestimated if these ocean regions are warming. For example, neglecting shallow waters is
discussed to account for more than 10% for 0–2000m OHC trends (Savita et al., 2022; von
Schuckmann et al., 2014), and about 4% for the Arctic area (Mayer et al., 2021). The assessment
is based on three distinct periods to account for the evolution of the observing system, i.e., 1960–
2020 (i.e., "historical"), 1993–2020 (i.e., "altimeter era") and 2006–2020 (i.e., "golden Argo era").
All time series go up to 2020 – which was one of the principal limitations for the inclusion of some
products. Our final estimates of OHC for the 0-300m, 0-700m, 700-2000m and 0-2000 m depth
layers are the ensemble average of all products, with the uncertainty range defined by the standard
deviation ($2\sigma$) of the corresponding estimates used (Fig. 2).
For the trend evaluation we have followed the most recent study of (Cheng et al., 2022), and used
a Locally Weighted Scatterplot Smoothing (LOWESS) approach to reduce the effect of high-
frequency variability (e.g., year-to-year variability), data noise or changes in the GCOS as it relies
on a weighted regression (Cleveland, 1979) within a prescribed span width of 25 years for the
historical and altimeter era, and 15 years for the recent period 2006-2020. The change in OHC(t)
over a specific period, $\Delta$OHC, is then calculated by subtracting the first value to the last value of
the fitted time series, $OHC_{LOWESS(t)}$, to obtain the trend while dividing by the considered period.
To obtain an uncertainty range on the trend estimate, and take into account the sensitivity of the
calculation to interannual variability, we implement a Monte-Carlo simulation to generate 1000
surrogate series $OHC_{random}(t)$, under the assumption of a given mean (our "true" time series
OHC(t)) (Cheng et al., 2022). Each surrogate $OHC_{random}(t)$ consists of the "true" time serie OHC(t)
plus a randomly generated residual which follows a normal (Gaussian) distribution, and which is
included in an envelope equal to 2 times the uncertainty associated to the time series. Then, a
LOWESS fitted line is estimated for each of the 1000 surrogates. The 95% confidence interval for
the trend is then calculated based on $\pm$ 2 times the standard deviation ($\pm$ 2-$\sigma$) of all 1000 trends of
the surrogates. However, the use of either trend estimates following a linear, or LOWESS
approach, or the approach discussed in (Palmer et al., 2021) lead to consistent results within
uncertainties (not shown).
In agreement with (Cheng et al., 2019; Gulev et al., 2021), our results reveal a continuous increase
of ocean warming over the entire study period (Fig. 2). Moreover, rates of global ocean warming
have increased over the 3 different study periods, i.e., historical up to the recent decadal change.
The trend values are all given in Table 1. The major fraction of heat is stored in the upper ocean
(0–300 m and 0–700 m depth). However, heat storage at intermediate depth (700–2000 m)
increases at a nearly comparable rate as reported for the 0–300 m depth layer (Table 1, Fig. 3).
There is a general agreement among the 16 international OHC estimates (Fig. 3). However, for
some periods and depth layers the standard deviation reaches maxima to about 0.3 W m$^{-2}$. All
products agree on the fact that global ocean warming rates have increased in the past decades and
doubled since the beginning of the altimeter era (1993–2020 compared with 1960–2020) (Fig. 3).
Moreover, there is a clear indication that heat sequestration into the deeper ocean layers below
700 m depth took place over the past 6 decades linked to an increase in OHC trends over time (Fig.





3). Ocean warming rates for the 0–2000 m depth layer reached record rates of 1.0 (0.7) ±0.3 W m$^{-2}$
over the period 2006-2020 for the ocean (global) area.

|  | Ocean Heat Content linear trends (W/m$^2$) | | | |
|---|---|---|---|---|
|  | **0-300m** | **0-700m** | **0-2000m** | **700-2000m** |
| **1960-2020** | 0.24 ± 0.1 | 0.34 ± 0.1 | 0.53 ± 0.1 | 0.18 ± 0.04 |
| **1971-2020** | 0.30 ± 0.1 | 0.44 ± 0.1 | 0.62 ± 0.1 | 0.21 ± 0.03 |
| **1993-2020** | 0.39 ± 0.1 | 0.60 ± 0.1 | 0.86 ± 0.2 | 0.30 ± 0.04 |
| **2006-2020** | 0.44 ± 0.1 | 0.64 ± 0.1 | 1.00 ± 0.3 | 0.38 ± 0.1 |

*Table 1: OHC trends using LOWESS (Locally Weighted Scatterplot Smoothing, see text for more*
*details) as derived from the ensemble mean (Fig. 2) for different time intervals, as well as different*
*integration depths. The regression was done for each time period (1960 - 2020, 1971 - 2020, 1993*
*- 2020, 2006 -2020). A time window of 25 years was used for the periods that allowed it (1960 -*
*2020, 1971 - 2020, 1993 - 2020). For the period 2006 - 2020, a time window of 15 years was used.*
*Note that values are given in Wm$^{-2}$ relative to the ocean surface area between 60°S and 60°N and*
*are limited to the 300 m bathymetry of each product. See also text and Fig. 2-3 for more details.*
For the deep OHC changes below 2000 m, we adapted an updated estimate from (Purkey &
Johnson, 2010) (PG10 hereinafter) from 1991 to 2020, which is a constant linear trend estimate
(0.97 ± 0.48 ZJ yr$^{-1}$, 0.06 ±0.03 W m$^{-2}$) derived from a global integration of OHC below 2000 m
using basin scale deep ocean temperature trends from repeated hydrographic sections. Some recent
studies strengthened the results in PG10 (Desbruyères et al., 2016; Zanna et al., 2019). Desbruyères
et al. (2016) examined the decadal change of the deep and abyssal OHC trends below 2000 m in
the 1990s and 2000s, suggesting that there has not been a significant change in the rate of decadal
global deep/abyssal warming from the 1990s to the 2000s and the overall deep ocean warming rate
is consistent with PG10. Using a Green's function method and ECCO reanalysis data, Zanna et al.
(2019) reported a deep ocean warming rate of ~0.06 W m$^{-2}$ during the 2000s, consistent with PG10
used in this study. Zanna et al. (2019) shows a fairly weak global trend during the 1990s, different
from observation-based estimates. This mismatch might come from how surface-deep connections
are represented in ECCO reanalysis data and the use of time-mean Green's functions in Zanna et
al. (2019), as well as from the sparse coverage of the observational network for relatively short
time spans. Furthermore, combining hydrographic and deep-Argo floats, a recent study (Johnson
et al., 2019) reported an accelerated warming in the South Pacific Ocean in recent years, but a
global estimate of the OHC rate of change over time is not available yet, and the rates of warming
may vary by ocean basin.
Before 1990, we assume zero OHC trend below 2000 m due to insufficient global observations
below 2000m, following the methodology in some studies (Cheng et al. 2017; 2022), IPCC-AR5
(Rhein et al., 2013) and IPCC-AR6 (Forster et al., 2022; Gulev et al. 2021). The deep warming is
likely driven by decadal variability in deep water formation rates, which could have been in a non-
steady state mode prior to 1990, introducing additional uncertainty to the pre-1990 OHC estimates.
Using surface temperature observations and assuming the heat is advected by mean circulation,



Zanna et al. (2019) shows a near-zero (small cooling trend) OHC trend below 2000 m from the
1960s to 1980s, suggesting the assumption of zero-trend before 1990 might be small. The derived
time following PG10 series after 1991 and zero-trend before 1990 is used for the Earth energy
inventory in Sect. 5. A centralized (around the year 2006) uncertainty approach has been applied
for the deep (>2000 m depth) OHC estimate following the method of Cheng et al. (2017), which
allows us to extract an uncertainty range over the period 1993–2018 within the given [lower (0.96–
0.48 ZJ yr$^{-1}$), upper (0.96+0.48 ZJ yr$^{-1}$)] range of the deep OHC trend estimate. We then extend
the obtained uncertainty estimate back from 1993 to 1960, with 0 OHC anomaly.

## 3. Heat available to warm the atmosphere

The heat content of the atmosphere is small in absolute terms, since its heat capacity as a gas is
small compared to the one of the other Earth subsystems discussed in this paper. Yet it is by no
means negligible, since in relative terms, the atmospheric heat gain is rapid over the recent decades
and has a high impact on human life (Fig. 1). As for Earth's surface, widespread and rapid changes
are ongoing in the atmosphere due to human-induced climate change (IPCC, 2021).
Atmospheric observations show a warming of the troposphere and a cooling and contraction of the
stratosphere since at least 1979 (Pisoft et al., 2021; Steiner et al., 2020). In the tropics, the upper
troposphere has warmed faster than the near-surface atmosphere since at least 2001, as seen with
the new observation technique of GPS radio occultation (Gulev et al., 2021; Steiner et al., 2020a;
2020b), while observations based on microwave soundings have likely underestimated
tropospheric temperature trends in the past (Santer et al., 2021; Zou et al., 2021).
Recently, a continuous rise of the tropopause has been observed for 1980 to 2020 over the northern
hemisphere (Meng et al., 2022). The increase is equally due to tropospheric warming and
stratospheric cooling in the period 1980 to 2000 while the rise after 2000 resulted primarily from
enhanced tropospheric heat gain. Moreover, indications exist on a widening of the tropical belt (Fu
et al., 2019; Grise et al., 2019; Staten et al., 2020) as well as on changes in the seasonal cycle
(Santer et al., 2022). However, changes in atmospheric circulation and conditions for extreme
weather are still subject to uncertainty (Cohen et al., 2020) while the occurrence of heat-related
extreme weather events has clearly increased over the recent decades (Cohen et al., 2020; IPCC,
2021), with high risks for society, economy, and the environment (Fischer et al., 2021).
A regular assessment of atmospheric heat content changes is hence critical for a complete overview
of energy and mass exchanges with other climate components and for a complete energy budgeting
of Earth's climate system.

### 3.1 Atmospheric heat content

In a globally averaged and vertically integrated sense, heat accumulation in the atmosphere arises
from a small imbalance between net energy fluxes at the top-of-atmosphere (TOA) and the surface
(denoted $s$). The heat energy budget of the vertically integrated and globally averaged atmosphere
(indicated by the global averaging operator $<.>$) reads as follows (Mayer et al., 2017):



$$\frac{\partial AE}{\partial t} > N_{TOA} > -F_s > -F_{snow} > -F_{PE} >, (1)$$

where the vertically integrated atmospheric energy content $AE$ per unit surface area [Jm$^{-2}$] reads
$$AE = \int_{z_s}^{z_{TOA}} \rho \left( c_v T + g(z - z_s) + L_e q + \frac{1}{2}V^2 \right) dz. (2)$$

In Equation (1), formulated in mean-sea-level altitude ($z$) coordinates used here for integrating
over observational data, $N_{TOA}$ is the net radiation at top of the atmosphere, $F_s$ is the net surface
energy flux defined as the sum of net surface radiation and latent and sensible heat fluxes, $F_{snow}$
denotes the latent heat flux associated with snowfall, and $F_{PE}$ additionally accounts for sensible
heat of precipitation. See Mayer et al. (2017) or von Schuckmann et al. (2020) for a discussion of
the latter two terms, which are small on a global scale and hence often neglected.
Equation (2) provides a decomposition of $AE$ into sensible heat energy (sum of the first two terms,
internal heat energy and gravity potential energy), latent heat energy (third term), and kinetic
energy (fourth term), where $\rho$ is the air density, $c_v$ the specific heat for moist air at constant volume,
$T$ the air temperature, $g$ the acceleration of gravity, $L_e$ the temperature-dependent effective latent
heat of condensation $L_v$ or sublimation $L_s$ (the latter relevant below 0 °C), $q$ the specific humidity
of the moist air, and $V$ the wind speed. We neglect atmospheric liquid water droplets and ice
particles as separate species, as their amounts and especially their trends are small.
In computing $AE$ for the purpose of this update to the von Schuckmann et al. (2020) heat storage
assessment, we continued to use the formulations described therein, including that we refer to the
(geographically aggregated) $AE$ as atmospheric heat content (AHC) in this context, acknowledging
the dominance of the heat-related terms in Eq. (2). Briefly, in deriving the AHC from observational
datasets, we accounted for the intrinsic temperature-dependence of the latent heat of water vapor
in formulating $L_e$ (for details see Gorfer, 2022) while the reanalysis derivations approximated $L_e$
by constant values of $L_v$, as this simplification is typically also made in the assimilating models
(e.g., ECMWF-IFS, 2015). As another small difference, the observational estimations neglected
the kinetic energy term in Eq. (2) while the reanalysis estimations accounted for it. The resulting
differences in AHC anomalies from any of these differences are negligibly small, however,
especially when considering trends over time.
**3.2 Datasets and heat content estimation**
Turning to the actual datasets used, the AHC and its changes and trends over time can be quantified
using various data sources, observation-based and reanalyses. Reassessing possible data sources,
we extended the high-quality datasets that we used in the initial von Schuckmann et al. (2020)
assessment. In particular, we updated the time period from 2018 to 2020 and improved the back-
extension from 1980 to 1960. Specifically, the adopted datasets and the related AHC data record
preparations can be summarized as follows.
Atmospheric reanalyses combine observational information from various sources (radiosondes,
satellites, weather stations, etc.) and a dynamical model in a statistically optimal way. These data
have reached a high level of maturity, thanks to continuous improvement work since the early
1990s (Hersbach et al., 2018). Especially reanalyzed thermodynamic state variables, like



temperature and water vapor that are most relevant for AHC computation, are of high quality and
suitable for climate studies, although temporal discontinuities introduced from changing observing
systems continue to deserve due attention (Berrisford et al., 2011; Chiodo & Haimberger, 2010;
Hersbach et al., 2020; Mayer et al., 2021).
We use the latest generation of reanalyses, including ECMWF's Fifth generation reanalysis ERA5
(Bell et al., 2021; Hersbach et al., 2020), JMA's reanalysis JRA55 (Kobayashi et al., 2015), and
NASA's Modern-Era Retrospective analysis for Research and Applications version 2 (MERRA2)
(Gelaro et al., 2017). ERA5 and JRA55 are both available over the full joint timeframe of this heat
storage assessment from 1960 to 2020, while MERRA2 complements these from 1980 to 2020.
The additional JRA55C reanalysis variant of JRA55, included for initial inter-comparison in von
Schuckmann et al. (2020), is no longer used since it is available to 2012 only and due to its
similarity to JRA55 is not adding appreciable complementary value.
In addition to these three reanalyses, the datasets from two climate-quality observation techniques
are used, for complementary observational AHC estimates. These include the Wegener Center
(WEGC) multi-satellite radio occultation (RO) data record, WEGC OPSv5.6 (Angerer et al., 2017;
Steiner et al., 2020b), over 2002-2020 and its radiosonde (RS) data record derived from the high-
quality Vaisala sondes RS80/RS92/VS41, WEGC Vaisala (Ladstädter et al., 2015), covering 1996-
2020. These RO and RS data sets provide atmospheric profiles of temperature, specific humidity,
and density that are vertically completed by collocated ERA5 profiles in domains not fully covered
by the data (e.g., in the lower troposphere for RO or at polar latitudes for RS). Similar to dropping
the JRA55C reanalysis variant for no longer adding value, the microwave sounding unit (MSU)
observational data, inter-compared in von Schuckmann et al. (2020), are no longer used.
From the observational data, the AHC is estimated by first evaluating Eq. (2) (using all terms for
total and the third term only for latent AHC) at each available profile location and subsequently
deriving it as volumetric heat content, for up to global scale, from vertical integration, temporal
averaging, and geographic aggregation according to the approach summarized in von Schuckmann
et al. (2020) and described in detail by (Gorfer, 2022). For the reanalyses, the estimation is based
on the full gridded fields. Applying the approach for crosscheck to reanalysis profiles sub-sampled
at observation locations only, confirms its validity as it accurately leads to the same AHC results
as from the full gridded fields.
**3.3 Atmospheric heat content change since 1960 and its amplification**
Figure 4 shows the resulting global AHC change inventory over 1960 to 2020 (61 years record),
in terms of total AHC anomalies for each data type (Fig. 4a), and for the ensemble mean with
trends for selected periods and uncertainty estimates (Fig. 4c). The selected trend periods align
with those for ocean data and with availability of atmospheric data sets (see subsection 3.2 above)
and represent a reference trend 1961-2000 plus recent trends of the last about 30, 20, and 15 years,
respectively. Latent AHC anomalies, a key component of the AHC (Matthews et al., 2022), are
also shown (Fig. 4b and 4d). Compared to von Schuckmann et al. (2020), the AHC data have the
ENSO signal removed (with ENSO regressed out via the Nino 3.4 Index; and cross-check with
non-ENSO-corrected data showing that trend differences are reasonably small). Variability due to





volcanic eruptions is still included, however, and may somewhat influence the trends over 1993-
2020, which start in the cold anomaly after the Pinatubo eruption (Santer et al., 2001).
The latent AHC (Fig. 4b and 4d), which accounts for about one-quarter of the total AHC, exhibits
a qualitatively similar temporal evolution as total AHC, however with larger relative uncertainty
compared to the total AHC. The RO and RS data sets in Fig. 3b show some differences, particularly
the low latent AHC values in the 1990s and early 2000s from the RS WEGC Vaisala data set likely
stem from known dry biases of the RS80/RS90/RS92 humidity sensors (Verver et al., 2006; Vömel
et al., 2007). Estimated trends based on these RS data are thus likely too high, although the overall
increase in latent AHC is substantial also in the other datasets.

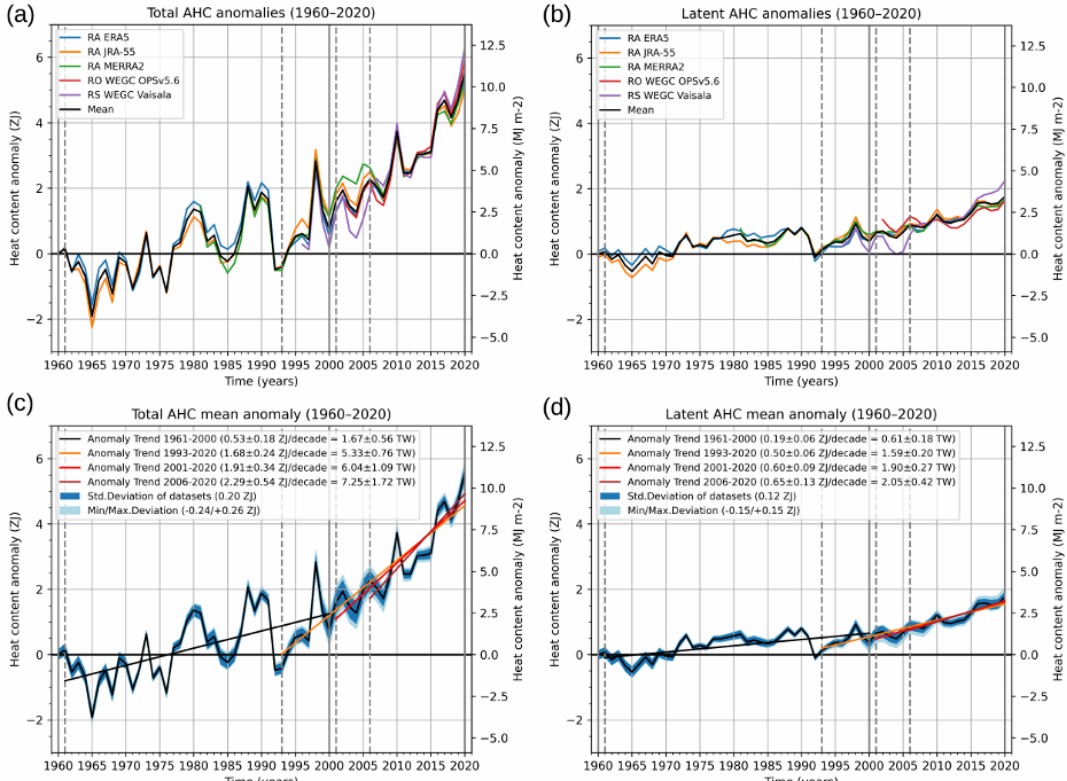

**Figure 4.** *Annual-mean global AHC anomalies from 1960 to 2020 of total AHC (left) and latent-*
*only AHC (right), respectively, of three different reanalyses and two different observational*
*datasets shown together with their mean (top), and the mean AHC anomaly shown together with*
*four representative AHC trends and ensemble spread measures of its underlying datasets (bottom).*
*The in-panel legends identify the individual datasets (top) and the selected trend periods together*
*with the associated trend values (plus 90 % confidence range) and ensemble spread measures*
*(bottom), the latter including the time-average standard deviation and minimum/maximum*
*deviations of the individual datasets from the mean.*

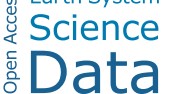


The results clearly show that the AHC trends have increased from the earlier decades represented
by the 1961-2000 trend of near 1.7 TW. We find the mean trend about 2.5 times higher over 1993-
2020 (about 5.3 TW) and about four times higher in the most recent two decades (about 6-7 TW),
a period that is already covered also by the RO and RS records. Latent AHC trends in the most
recent periods are 3 times larger than the 1961-2000 reference period. Since 1971, the heat gain in
the atmosphere amounts to $5 \pm 1$ ZJ (see also Fig. 8).
The remarkable amplification of total AHC and latent AHC trends is highlighted in Figure 4 and
summarized in Table 2 for the representative recent periods vs. the 1961-2000 reference period.
The 1961-2000 and 1993-2020 periods were covered by reanalysis only, while the WEGC Vaisalä
RS dataset additionally covers the 2001-2020 and 2006-2020 periods and the RO dataset the most
recent period (see dataset descriptions in subsection 3.2). The larger diversity of recent datasets
induces more spread; for example, the RS dataset shows an amplification factor of near 4.5 in the
global total AHC gain for 2001-2020, while the amplification factors from the reanalyses range
from 2.6 to 3.8. Amplifications are generally largest in the southern hemisphere extratropics and
weakest in the tropics. In the most recent period 2006-2020, the amplification factors are strongest,
with the RS and RO data sets on the high end of the spread (near factor 5 in global total AHC) and
somewhat smaller but still high from the reanalyses (around factor 4).

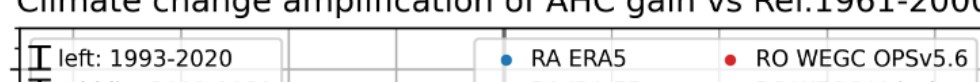

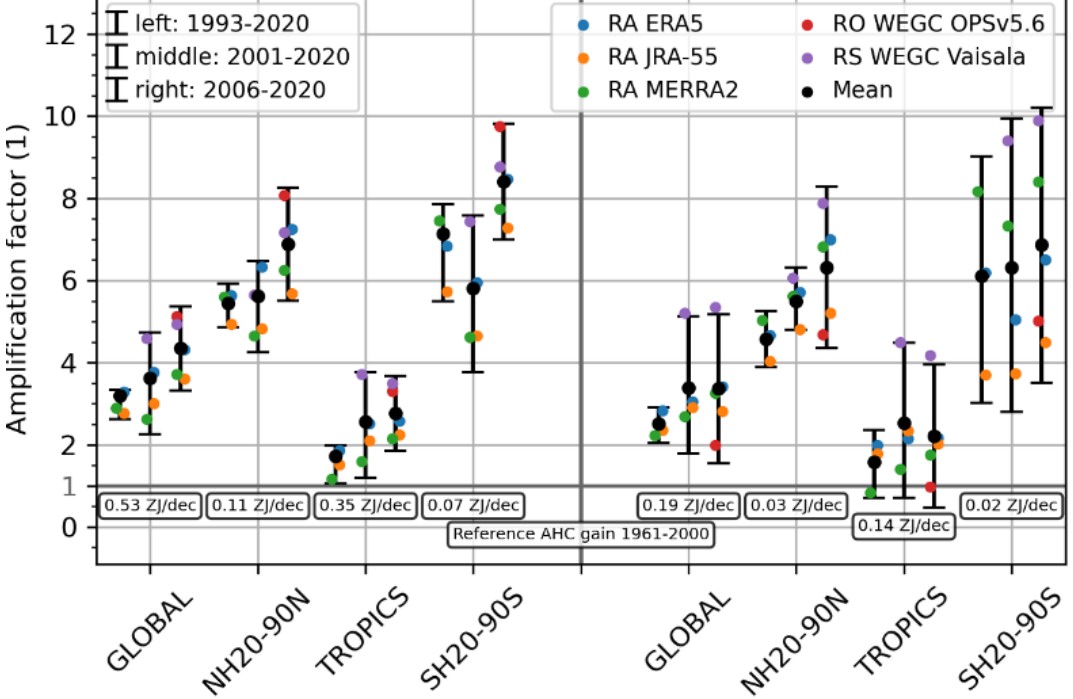




**Figure 5.** *Amplification of long-term trends in AHC anomalies ("AHC gain") for total AHC (left)*
*and latent-only AHC (right) in four geographic domains (global, northern-hemisphere*
*extratropics, tropics, southern-hemisphere extratropics) for three recent time periods (legend*
*upper-left) expressed as a ratio of the trend of each period relative to the trend in the previous-*
*century reference period 1961-2000 (noted below the "amplification factor = 1" reference line).*
*The amplification factor for each recent-trend case (for the four domains of both total and latent*
*AHC) is depicted for the mean anomaly serving as best estimate (larger black circles), the related*
*recent trends in the individual-dataset anomalies (colored circles as per upper-right legend). The*
*related 90 % uncertainty range (black "error bar") is estimated from the spread (standard*
*deviation) of the individual-dataset amplification factors. The trend in the mean anomaly over*
*1961-2000 is used as the reference AHC gain.*
For the latent AHC amplification factors, we see moderate values in the 1993-2020 period in the
global mean and tropics. In the tropics, the lower uncertainty bound for amplification is slightly
below 1 during all three recent trend periods. The spread of the amplification factors increases for
the most recent periods, which is on the one hand due to the shorter period duration. The range
increase is also related to the introduction of the RS and RO data sets after 1993-2020 which
contribute the largest and smallest latent AHC gain amplification factors. For 2006-2020, the
global mean amplification factor from RO is about 2, whereas from the RS data set it is near 5.
Regarding latitudinal bands, the amplification factors are again strongest in the extratropics,
exhibiting a large spread especially in the southern extratropics. The relatively large amplification
factors of the RS WEGC Vaisala data set are likely exaggerated due to the well documented dry
bias of the early RS humidity sensors as noted above (Vömel, 2007; Verver et al., 2006).
Despite the uncertainties and spread described, the overall message from Figure 5 and Table 2 is
very clear and substantially reinforcing the evidence from the initial von Schuckmann et al. (2020)
assessment: the trends in the AHC, including in its latent heat component, show that atmospheric
heat gain has accelerated over the recent decades at an unprecedented rate.

| Domain | Time range | Total AHC Gain | | Latent AHC Gain | |
|---|---|---|---|---|---|
| | | Gain ZJ/decade (TW) | Amplification vs Ref. | Gain ZJ/decade (TW) | Amplification vs Ref. |
| **GLOBAL** | 1993-2020 | 1.68±0.24 (5.33±0.76) | 3.19 [2.63 to 3.34] | 0.50±0.06 (1.59±0.20) | 2.51 [2.05 to 2.91] |
| | 2001-2020 | 1.91±0.34 (6.04±1.09) | 3.62 [2.27 to 4.73] | 0.60±0.09 (1.90±0.27) | 3.39 [1.79 to 5.13] |
| | 2006-2020 | 2.29±0.54 (7.25±1.72) | 4.35 [3.33 to 5.36] | 0.65±0.13 (2.05±0.42) | 3.37 [1.55 to 5.18] |
| | Ref. 1961-2000 | 0.53±0.18 (1.67±0.56) | 1.0 | 0.19±0.06 (0.61±0.18) | 1.0 |
| **NH20-90N** | 1993-2020 | 0.62±0.11 (1.97±0.35) | 5.44 [4.86 to 5.92] | 0.16±0.02 (0.50±0.08) | 4.57 [3.90 to 5.26] |
| | 2001-2020 | 0.64±0.15 (2.03±0.47) | 5.62 [4.26 to 6.48] | 0.18±0.03 (0.58±0.11) | 5.50 [4.79 to 6.31] |
| | 2006-2020 | 0.79±0.25 (2.49±0.80) | 6.89 [5.51 to 8.26] | 0.22±0.05 (0.70±0.17) | 6.32 [4.36 to 8.28] |
| | Ref. 1961-2000 | 0.11±0.08 (0.36±0.24) | 1.0 | 0.03±0.02 (0.11±0.05) | 1.0 |
| **TROPICS** | 1993-2020 | 0.60±0.13 (1.90±0.41) | 1.72 [1.05 to 1.98] | 0.24±0.04 (0.75±0.12) | 1.58 [0.71 to 2.36] |
| | 2001-2020 | 0.89±0.15 (2.82±0.47) | 2.56 [1.20 to 3.77] | 0.31±0.05 (1.00±0.16) | 2.52 [0.70 to 4.49] |
| | 2006-2020 | 0.96±0.24 (3.04±0.77) | 2.76 [1.86 to 3.67] | 0.31±0.07 (0.99±0.22) | 2.22 [0.48 to 3.96] |
| | Ref. 1961-2000 | 0.35±0.08 (1.10±0.25) | 1.0 | 0.14±0.03 (0.45±0.11) | 1.0 |
| **SH20-90S** | 1993-2020 | 0.46±0.09 (1.46±0.29) | 7.14 [5.49 to 7.86] | 0.11±0.02 (0.33±0.05) | 6.11 [3.02 to 9.02] |
| | 2001-2020 | 0.37±0.17 (1.18±0.52) | 5.80 [3.76 to 7.58] | 0.10±0.03 (0.32±0.08) | 6.31 [2.81 to 9.95] |
| | 2006-2020 | 0.54±0.25 (1.71±0.79) | 8.40 [6.99 to 9.81] | 0.11±0.04 (0.36±0.12) | 6.87 [3.52 to 10.22] |
| | Ref. 1961-2000 | 0.06±0.06 (0.20±0.18) | 1.0 | 0.02±0.01 (0.05±0.05) | 1.0 |






***Table 2.*** *Long-term trend values in mean AHC anomalies (AHC gains; in units ZJ/decade and TW)*
*and amplification factors vs. the 1961-2000 reference gain (grey "Ref." lines), for total AHC (left*
*block) and latent-only AHC (right block) for the three recent time periods in four geographic*
*domains as illustrated in Figure 4. The AHC gain and amplification values are listed together with*
*their 90% confidence ranges.*


## 4. Heat available to warm land

In previous studies the land term of the Earth heat inventory was considered as the heat used to
warm the continental subsurface (Hansen et al. 2011; Rhein et al. 2013; von Schuckmann et al.
2020). Temperature changes within the continental subsurface are typically retrieved by analyzing
the global network of temperature-depth profiles, measured mostly in the northern hemisphere,
southern Africa, and Australia. Each temperature profile records changes in subsurface
temperatures caused by the heat propagated through the ground due to alterations in the surface
energy balance (Cuesta-Valero et al., 2022a). Such perturbations in the subsurface temperature
profiles can be analyzed to recover the changes in past surface conditions that generated the
measured profile, allowing a reconstruction of the evolution of ground surface temperatures and
ground heat fluxes at decadal to centennial time scales (Beltrami et al., 2002; Beltrami &
Mareschal, 1992; Demezhko & Gornostaeva, 2015; Hartmann & Rath, 2005; Hopcroft et al., 2007;
Jaume-Santero et al., 2016; Lane, 1923; Pickler et al., 2016; Shen et al., 1992). Although previous
estimates only considered changes in ground temperatures for representing the heat storage by
exposed land, ground heat storage has been found to be the second largest term of the Earth heat
inventory accounting for 4 % to 6 % of the total heat in the Earth System (von Schuckmann et al.
2020, section 6).

The ground heat is, nevertheless, not the only energy component of the continental landmasses.
Other processes with large thermodynamic coefficients, such as permafrost thawing and the
warming of inland water bodies, occur across large areas, leading to the exchange of large amounts
of heat with their surroundings over time. To account for those heat exchanges, a recent study
(Cuesta-Valero et al., 2022a) has estimated the heat uptake by permafrost thawing and the warming
of inland water bodies, as well as ground heat storage from subsurface temperature profiles,
resulting in a comprehensive estimate of continental heat storage. The authors used the same global
network of subsurface temperature profiles as in von Schuckmann et al. (2020) to estimate ground
heat storage but applied an improved inversion technique to analyze the profiles. This new
technique is based on combining bootstrapping sampling with a widely-used Singular Value
Decomposition (SVD) algorithm (e.g., Beltrami et al., 1992) to retrieve past changes in surface
temperatures and ground heat fluxes, which also resulted in smaller uncertainty estimates for
global results (Cuesta-Valero et al., 2022b). Heat uptake from permafrost thawing was estimated
using a large ensemble of simulations performed with the CryoGridLite permafrost model
(Nitzbon et al., 2022). Ground stratigraphies required for this purpose, including ground ice
distributions, were generated using various global ground datasets. Latent heat storage due to
melting of ground ice is evaluated to a depth of 550 m over the Arctic region. Uncertainty ranges
are evaluated using 100 parameter ensemble simulations with strongly varied soil properties and
soil ice distributions. The climate forcing at the surface is based on a paleoclimate simulation



Data

performed by the Commonwealth Scientific and Industrial Research Organization (CSIRO)
providing the initialization of the permafrost model, and data from the ERA-Interim reanalysis
since 1979 onwards. Heat storage by inland water bodies was estimated by integrating water
temperature anomalies in natural lakes and reservoirs from a set of Earth System Model (ESM)
simulations participating in the Inter-Sectoral Impact Model Intercomparison Project phase 2b
(ISIMP2b) (Frieler et al., 2017; Golub et al., 2022; Grant et al., 2021). Heat storage is then
computed using simulations with four global lake models following the methodology presented in
(Vanderkelen et al., 2020), but replacing the cylindrical lake assumption in that study for a more
detailed lake morphometry, which leads to a more realistic representation of lake volume.

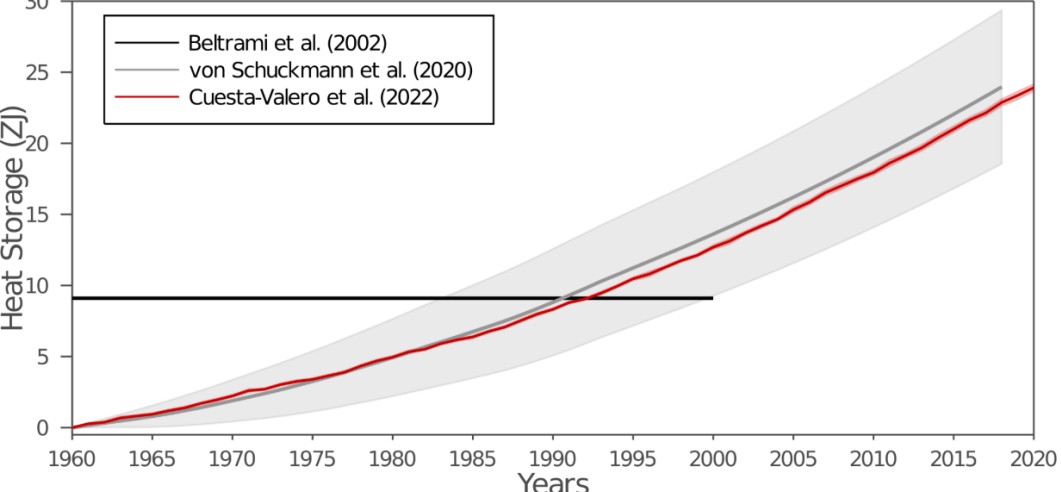

*Figure 6: Continental heat storage from Beltrami et al. (2002) (black), von Schuckmann et al.*
*(2020) (gray), and Cuesta-Valero et al. (2022a) (red). Gray and red shadows show the uncertainty*
*range of the heat storage from von Schuckmann et al. (2020) and Cuesta-Valero et al. (2022a),*
*respectively.*
Figure 6 shows the three main estimates of heat gain by the continental landmasses since 1960.
The first global estimate of continental heat storage was provided by Beltrami et al. (2002),
consisting of changes in ground heat content for the period 1500-2000 as time steps of 50 years
(black line in Figure 6). These estimates were retrieved by inverting 616 subsurface temperature
profiles constituting the global network of subsurface temperature profiles in 2002, yielding a heat
gain of 9.1 ZJ during the second half of the 20th century. A comprehensive update was included
in von Schuckmann et al. (2020) using the results of (Cuesta-Valero et al., 2021) (gray line in
Figure 6), with the main difference consisting in the use of a larger dataset with 1079 subsurface
temperature profiles. Since many of these new profiles were measured at a later year than those in
Beltrami et al. (2002), the inversions from this new data set were able to include the recent
warming of the continental subsurface, yielding higher ground heat content than those from
Beltrami et al. (2002). Concretely, the estimates in von Schuckmann et al. (2020) showed a heat
gain of 24 ± 5 ZJ from 1960 to 2018.



Recently, a new estimate of continental heat gain including the heat used in permafrost thawing
and in warming inland water bodies was presented in Cuesta-Valero et al. (2022a) (red line in
Figure 6), achieving a heat gain of 24 ± 1 ZJ since 1960, and 22 ± 1 ZJ since 1971 (see also Fig.
8). Although this estimate uses the same 1079 measurement sites as in von Schuckmann et al.
(2020) and includes inland water bodies and permafrost thawing, it yields similar values of heat
content to those in von Schuckmann et al. (2020). These similar results are caused by the different
aggregation techniques used to derive the change in global ground heat storage in von Schuckmann
et al. (2020) and in Cuesta-Valero et al. (2022a). There is a difference of ~ 3 ZJ between the
average ground heat storage in Cuesta-Valero et al. (2022a) (21.6 ± 0.2 ZJ) and in von Schuckmann
et al. (2020) (24 ± 5 ZJ), which is similar to the heat storage in inland water bodies and the heat
storage due to permafrost thawing together (see below). Another important result is the narrower
confidence interval in estimates from Cuesta-Valero et al. (2022a), which is directly related to the
new bootstrap technique used to invert the subsurface temperature profiles (Cuesta-Valero et al.,
2022b). Heat storage within inland water bodies has reached 0.2 ± 0.4 ZJ since 1960, with
permafrost thawing accounting for 2 ± 2 ZJ. Therefore, ground heat storage is the main contributor
to continental heat storage (90 %), with inland water bodies accounting for 0.7 % of the total heat,
and permafrost thawing accounting for 9 %. Despite the smaller proportion of heat stored in inland
water bodies and permafrost thawing, several important processes affecting both society and
ecosystems depend on the warming of lakes and reservoirs, and on the thawing of ground ice
(Gädeke et al., 2021). Therefore, it is important to continue quantifying and monitoring the
evolution of heat storage in all three components of the continental landmasses.
**5. Heat utilized to melt ice**
Changes in Earth's cryosphere affect almost all other elements of the environment including the
global sea level, ocean currents, marine ecosystems, atmospheric circulation, weather patterns,
freshwater resources and the planetary albedo (Abram et al., 2019). The cryosphere includes frozen
components of the Earth system that are at or below the land and ocean surface: snow, glaciers,
ice sheets, ice shelves, icebergs, sea ice, lake ice, river ice, permafrost and seasonally frozen
ground (IPCC, 2019). In this study, we estimate the heat uptake by the melting of ice sheets
(including both floating and grounded ice), glaciers and sea ice at global scale (Fig. 7).
Notwithstanding the important role snow cover plays in the Earth's energy surface budget as a
result of changes in the albedo (de Vrese et al., 2021; Qu & Hall, 2007; Weihs et al., 2021), or its
influence on the temperature of underlying permafrost (Jan & Painter, 2020; Park et al., 2015), or
on sea ice in the Arctic (Perovich et al., 2017; Webster et al., 2021) and Antarctica (Eicken et al.,
1995; Nicolaus et al., 2021; Shen et al., 2022), estimates of changes in global snow cover are still
highly uncertain and not included in this inventory. However, they should be considered in future
estimates. Similarly, changes in lake ice cover (Grant et al., 2021) are not taken into account here
and warrant more attention in the future. Permafrost is accounted for in the land component (see
section 4).

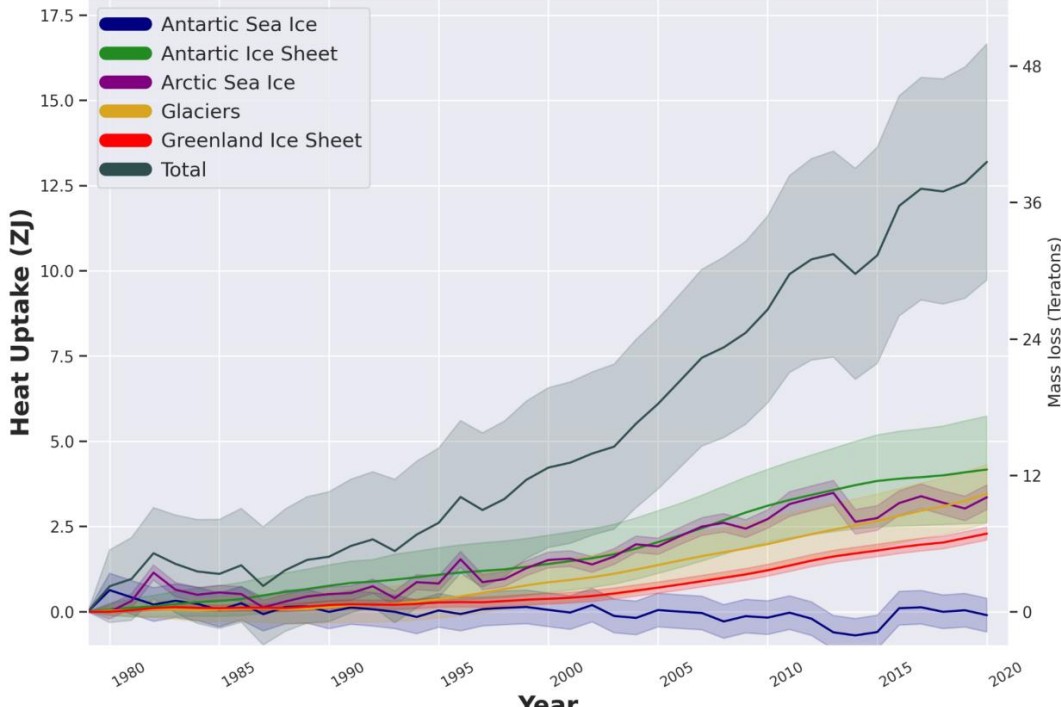

**Figure 7:** *Heat uptake (in ZJ) and Mass Loss (Trillions of tons) for the Antarctic Ice Sheet*
*(grounded and floating ice, green), Glaciers (orange), Arctic sea ice (purple), Greenland Ice Sheet*
*(grounded and floating ice, red) and Antarctic sea ice (blue), together with the sum of the energy*
*uptake within each one of its components (total, black). Uncertainties are 95% confidence*
*intervals provided as shaded areas, respectively. See text for more details.*
We equate the energy uptake by the cryosphere (glaciers, grounded and floating ice of the Antarctic
and Greenland Ice Sheets, and sea-ice) with the energy needed to drive the estimated mass loss. In
doing so we assume that the energy change associated with the temperature change of the
remaining ice is negligible. As a result, the energy uptake by the cryosphere is directly proportional
to the mass of melted ice:
$E = \Delta M * (L + c * \Delta T)$,
where, for any given component, $\Delta M$ is the mass of ice loss, L is the latent heat of fusion, c is the
specific heat capacity of the ice and $\Delta T$ is the rise in temperature needed to bring the ice to the
melting point. For consistency with previous estimates (Ciais et al., 2014; Slater et al., 2021; von
Schuckmann et al., 2020), we use a constant latent heat of fusion of $3.34 \times 10^5 \, \mathrm{J\,kg^{-1}}$, a specific
heat capacity of $2.01 \times 10^3 \, \mathrm{J/(kg\,^{\circ}C)}$ and, a density of ice of $917 \, \mathrm{kg/m^3}$. Estimating the energy used
to warm the ice to its melting point requires knowledge of the mean ice temperature for each
component. Here we assume a temperature of -15 °C for floating ice in Greenland, -2 °C for the
floating ice in Antarctica, -20 ± 10 °C for grounded ice in Antarctica and Greenland and 0 °C for



sea-ice and glaciers. Although this assumption is poorly constrained, the energy required to melt
ice is primarily associated with its phase transition and the fractional energy required for warming
is a small percentage ($< 1\%$ $°C^{-1}$) of the total energy uptake (Slater et al., 2021). Nevertheless, we
include an additional uncertainty of $\pm10$ °C on the assumed initial ice temperature within our
estimate of the energy uptake.
Grounded ice losses from the Greenland and Antarctic Ice Sheets from 1992 to 2020 are estimated
from a combination of 50 satellite-based estimates of ice sheet mass balance produced from
observations of changes in ice sheet volume, flow and gravitational attraction, compiled by the Ice
Sheet Mass Balance Intercomparison Exercise (IMBIE[8]) (Shepherd et al., 2018, 2019). To extend
those time-series further back in time, we use ice sheet mass balance estimates produced using the
input-output method, which combines estimates of solid ice discharge with surface mass balance
estimates. Satellite estimates of ice velocity are available from the Landsat historical archive from
1972 allowing the calculation of ice discharge before the 1990s while surface mass balance is
estimated from regional climate models. We extend the IMBIE mass balance time-series
backwards to 1979 for Greenland using (Mouginot et al., 2019) and (Mankoff et al., 2019) and for
Antarctica from 1972 to 1991 using (Rignot et al., 2019).
Changes in Antarctic floating ice shelves due to thinning between 1994 and 2017 are derived from
satellite altimetry reconstructions (Adusumilli et al., 2020). There were no estimates of ice shelf
thinning between 1979 and 1993, therefore we assume zero mass loss from ice shelf thinning
during that period. Changes in Antarctic ice shelves due to increased calving in the Antarctic
Peninsula and the Amundsen Sea sector are derived from ERS-1 radar altimetry (Adusumilli et al.
2020) for 1994-2017. For the 1979-1994 period, we only have data for changes in the extent of the
Antarctic Peninsula ice shelves from (Cook & Vaughan, 2010). These are converted to changes in
mass using an ice shelf thickness of 140 +/- 110 m ice equivalent which represents the range of
ice thickness values for the portions of Antarctic Peninsula ice shelves that have collapsed since
1994 (Adusumilli et al. 2020). Once icebergs calve off large Antarctic floating ice shelves, the
timescales of dissolution of the icebergs are largely unknown; therefore, we assumed a linear rate
of energy uptake between 1979–2018. For icebergs, we use an initial temperature of -16°C, which
was the mean ice temperature in the Ross Ice Shelf J-9 ice core (Clough & Hansen, 1979). There
are no large-scale observations or manifestations of significant firn layer temperature change for
the Antarctic ice shelf; for example, there is no significant trend in the observationally-constrained
model outputs of surface melt described in (Smith et al., 2020). Therefore, the change in
temperature of any ice that does not melt is assumed to be negligible.
Changes in the floating portions of the Greenland Ice Sheet include ice shelf collapse, ice shelf
thinning and tidewater glacier retreat. As in von Schuckmann et al. 2020, we assume no ice shelf
mass loss pre-1997 and estimate a loss of 13 Gt/yr post-1997 based on studies of Zacharie Isstrom,
C. H. Ostenfeld, Petermann, Jakobshavn, 79N and Ryder Glaciers (Moon & Joughin, 2008;
Motyka et al., 2011; Mouginot et al., 2015; Münchow et al., 2014; Wilson et al., 2017). We assign
a generous uncertainty of 50% to this value. For tidewater glacier retreat we note a mean retreat
rate of 37.6 m/yr during 1992-2000 and 141.7 m/yr during 2000-2010. We assume the former
estimate is also valid for 1979-1991 and the latter estimate is valid for 2011-2020. Assuming a

---

8      https://imbie.org



mean glacier width of 4 km and thickness of 400 m we estimate mass loss from glacier retreat to
be 9.3 Gt/yr during 1979-2000 and 35.1 Gt/yr during 2000-2020. Based on firn modeling we
assessed that warming of Greenland's firn has not yet contributed significantly to its energy uptake
(Ligtenberg et al., 2018).
The contributions from both the Antarctic and Greenland Ice Sheets to the EEI are obtained by
summing the mass loss from the individual components (ice shelf mass, grounded ice mass, and
ice shelf extent) for each ice sheet separately and, given that the datasets used for each component
are independent, the uncertainties were summed in quadrature. This is then converted to an energy
uptake according to the equation above.
Glaciers are another part of the land-based ice, and we here include glaciers found in the periphery
of Greenland and Antarctica, but distinct from the ice sheets, in our estimate. We build our estimate
on the international efforts to compile and reconcile measurements of glacier mass balance, under
the lead of the World Glacier Monitoring Service (WGMS[9]). Up to 2016, the results are based on
(Zemp et al., 2019), who combine geodetic mass balance observations from DEM differencing on
long temporal and large spatial scales with in-situ glaciological observations, which are spatially
less representative, but provide information of higher temporal resolution. Through this
combination, they achieve coverage that is globally complete yet retains the interannual variability
well. For 2017 to 2021, the numbers are based on the ad-hoc method of (Zemp et al., 2020), which
corrects for the spatial bias of the limited number of recent in-situ glaciological observations that
are available with short delay (WGMS, 2021), to derive globally representative estimates. Error
bars include uncertainties related to the in-situ and spaceborne observations, extrapolation to
unmeasured glaciers, density conversion, as well as to glacier area and its changes. For the
conversion from mass loss to energy uptake, only the latent heat uptake is considered, which is
based on the assumption of ice at the melting point, due to lack of glacier temperature data at the
global scale. Moreover, since the absolute mass change estimates are based on geodetic mass
balances, mass loss of ice below floatation is neglected. While this is a reasonable approximation
concerning the glacier contribution to sea-level rise, it implies a systematic underestimation of the
glacier heat uptake. While to our knowledge there are no quantitative estimates available of glacier
mass loss below sea level on the global scale, it is reasonable to assume that this effect is minor,
based on the volume-altitude distribution of glacier mass (Farinotti et al., 2019; Millan et al., 2022).
Further efforts are under way within the Glacier Mass Balance Intercomparison Exercise
(GlaMBIE[10]), particularly to reconcile global glacier mass changes including also estimates from
gravimetry and altimetry, and to further assess related sources of uncertainties (Zemp et al., 2019).
Sea ice, formed from freezing ocean water, and further thickened by snow accumulation is not
only another important aspect of the albedo effect (Kashiwase et al., 2017; R. Zhang et al., 2019)
and water formation processes (Moore et al., 2022), but also provides essential services for polar
ecosystems and human systems in the Arctic (Abram et al., 2019). Observations of sea-ice extent
are available over the satellite era, i.e. since the 1970s, but ice thickness data - required to obtain
changes in volume - have only recently become available through the launch of CryoSat-2 and
ICESat-2. For the Arctic, we use a combination of sea ice thickness estimates from from the Pan-

---

9    https://wgms.ch
10   https://glambie.org



Arctic Ice Ocean Modeling and Assimilation System (PIOMAS) between 1980 and 2011
(Schweiger et al., 2019; Zhang & Rothrock, 2003) and CryoSat-2 satellite radar altimeter
measurements between 2011 and 2020 when they are available (Slater et al., 2021; Tilling et al.,
2018). PIOMAS assimilates ice concentration and sea surface temperature data and is validated
with most available thickness data (from submarines, oceanographic moorings, and remote
sensing) and against multidecadal records constructed from satellite (Labe et al., 2018; Laxon et
al., 2013; Wang et al., 2016). We note that the PIOMAS domain does not extend sufficiently far
south to include all regions covered by sea ice in winter (Perovich et al., 2017). Given that the
entirety of the regions that are unaccounted for (e.g., the Sea of Okhotsk and the Gulf of St.
Lawrence) are only seasonally ice covered since the start of the record, this should not influence
the results. We convert monthly estimates of sea ice volume from CryoSat-2 satellite altimetry to
mass using densities of 882 and 916.7 kg/m$^3$ in regions of multi- and first-year ice respectively
(Tilling et al., 2018). During the summer months (May to September) the presence of melt ponds
on Arctic sea ice makes it difficult to discriminate between radar returns from leads and sea ice
floes, preventing the retrieval of summer sea ice thickness from radar altimetry (Tilling et al.,
2018). As a result, we use the winter-mean (October to April) mass trend across the Arctic for both
CryoSat-2 and PIOMAS estimates for consistency. According to PIOMAS, winter Arctic sea ice
mass estimates are 19 Gt/yr (6 %) smaller than the annual mass trend between 1980 and 2011 (-
324 Gt/yr) and so are a conservative estimate of Arctic sea ice mass change (Slater et al., 2021).
The uncertainty on monthly Arctic sea ice volume measurements from CryoSat-2 ranges from 14.5
% in October to 13 % in April (Slater et al., 2021; Tilling et al., 2018), and is estimated as ±1.8×10$^3$
km$^3$ for PIOMAS (Schweiger et al., 2011).
Satellite radar altimeter retrievals of sea ice thickness in the Southern Ocean are complicated by
the presence of thick snow layers with unknown radar backscatter properties on Antarctic sea ice
floes. As a result, no remote sensing estimates are available for Antarctic sea ice and we use sea
ice volume anomalies from the Global Ice-Ocean Modeling and Assimilation System (GIOMAS,
Zhang & Rothrock, 2003), the global equivalent to PIOMAS. GIOMAS output has been recently
validated against in-situ and satellite data by (Liao et al., 2022). We compute Antarctic sea ice
trends as annual averages between January and December. In the absence of a detailed
characterization of uncertainties for these estimates, we attribute the same uncertainty to GIOMAS
estimates as for PIOMAS (±1.8x10$^3$ km$^3$). For future updates of the GCOS Earth heat inventory,
we also aim to include observation-based (remote sensing) estimates in the Southern Ocean
(Lavergne et al., 2019).
Our estimate of the total heat gain in the cryosphere amounts to 14 ± 4 ZJ over the period 1971-
2020 (see also Fig. 8 and section 6), (assuming negligible contribution before 1979 according to
the data availability limitation), which is consistent with the estimate obtained in (von Schuckmann
et al., 2020) within uncertainties. Approximately half of the cryosphere's energy uptake is
associated with the melting of grounded ice, while the remaining half is associated with the melting
of floating ice (ice shelves in Antarctica and Greenland, Arctic sea ice). Compared to earlier
estimates, and in particular the 8.83 ZJ estimate from Ciais et al. (2013), this larger estimate is a
result both of the longer period of time considered and, also, the improved estimates of ice loss
across all components, especially the ice shelves in Antarctica. Contributions to the total
cryosphere heat gain are dominated by the Antarctic Ice Sheet (including the floating and grounded
ice, 33 ± 11%) and Arctic Sea ice (26 ± 3%), directly followed by the heat utilized to melt glaciers





(25 ± 7%). The Greenland Ice Sheet amounts to 17 ± 2%, whereas Antarctic sea ice is accounted
for with a non-significant contribution of 0.2 ± 4%.

### 6. The Earth heat inventory: where does the energy go?

Evaluations of the heat storage in the different Earth system components as performed in section
2-5 allow now for the establishment of the Earth heat inventory. Our results reconfirm a continuous
accumulation of heat in the Earth system since our estimate begins (Fig. 8). The total Earth system
heat gain in this study amounts to 380±62 ZJ over the period 1971–2020. For comparison, the heat
gain obtained in IPCC AR6 obtained a total heat gain of 434.9 [324.5 to 545.5] ZJ for the period
1971-2018, and is hence consistent with our estimate within uncertainties (Forster et al., 2021).
However, it is important to note that our estimate still excludes some aspects of Earth heat
accumulation, such as for example the shallow areas of the ocean. Although some estimates and
discussions have been provided to account for the relative contributions of these areas, these results
are still hampered by a number of assumptions and are challenging to be quantified with respect
to gaps in the observing system.

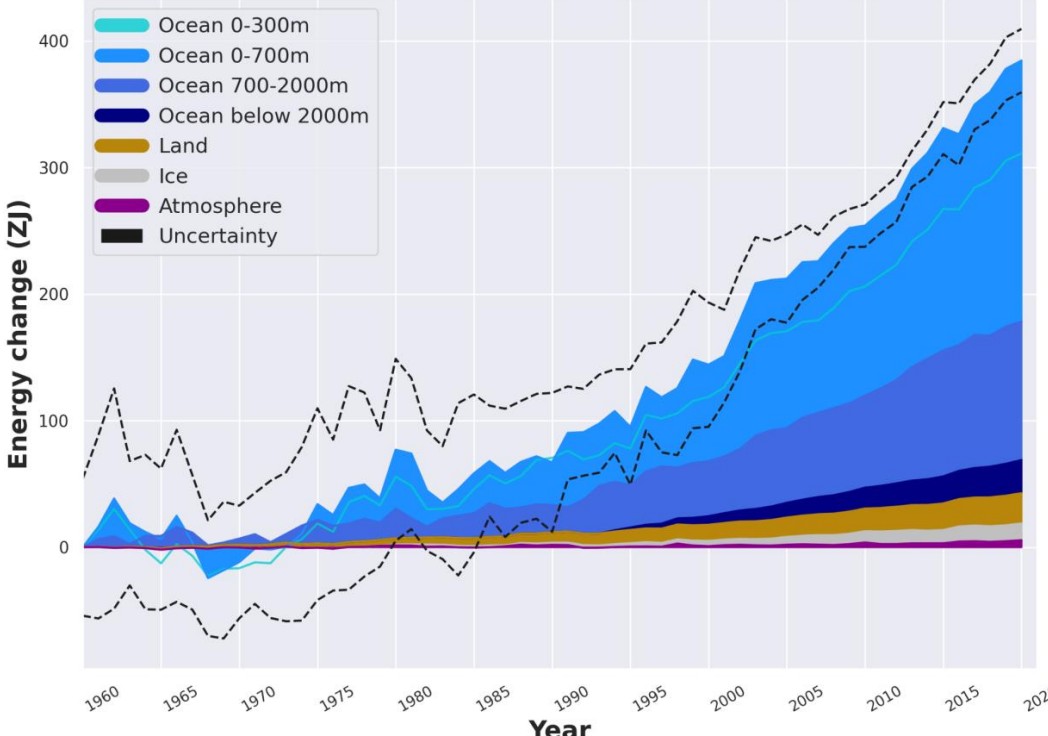

***Figure 8:*** *Total Earth system heat gain in ZJ (1 ZJ =$10^{21}$ J) relative to 1960 and from 1960 to*
*2020. The upper ocean (0–300 m, light blue line, and 0–700 m, light blue shading) accounts for*
*the largest amount of heat gain, together with the intermediate ocean (700–2000 m, blue shading)*
*and the deep ocean below 2000 m depth (dark blue shading). The second largest contributor is the*



*storage of heat on land (orange shading), followed by the gain of heat to melt grounded and*
*floating ice in the cryosphere (gray shading), and heating of the atmosphere (magenta shading).*
*Uncertainty in the ocean estimate also dominates the total uncertainty (dot-dashed lines derived*
*from the standard deviations ($2\sigma$) for the ocean, cryosphere, land and atmosphere). See sections*
*2-5 for more details of the different estimates. The dataset for the Earth heat inventory is published*
*at the German Climate Computing Centre (DKRZ, https://www.dkrz.de/) (see section 7).*
*Consistent with von Schuckmann et al. (2020), we obtain a total heat gain of 380±62 ZJ over the*
*period 1971–2020, which is equivalent to a heating rate (i.e., the EEI) of 0.48±0.1 W m$^{-2}$ applied*
*continuously over the surface area of the Earth (5.10×10$^{14}$ m$^2$). The corresponding EEI over the*
*period 2006–2020 amounts to 0.76±0.2 W m$^{-2}$. The LOWESS method and associated uncertainty*
*evaluations have been used as described in section 2.*
The estimate of heat storage in all Earth system components not only allows for obtaining a
measure of how much and where heat is available for inducing changes in the Earth system (Fig.
1), but also to improve the accuracy of the Earth's system total heat gain. In 1971-2020 and for the
total heat gain, the ocean accounts for the largest contributor with a $89 \pm 17\%$ fraction of the global
inventory. The second largest component in the Earth heat inventory relies on heat stored in land
with a $6 \pm 0.1\%$ contribution. The cryosphere component accounts for $4 \pm 1\%$, and the atmosphere
$1 \pm 0.2\%$. For the most recent era of best available GCOS data for the Earth heat inventory since
the year 2006, the fractions amount to $89 \pm 20\%$ for the ocean, $5 \pm 1\%$ for land, $4 \pm 3\%$ for the
cryosphere, and $2 \pm 0.4\%$ for the atmosphere.
The change of the Earth heat inventory over time allows for an estimate of the absolute value of
the Earth energy imbalance. Our results of the total heat gain in the Earth system over the period
1971-2020 is equivalent to a heating rate of 0.48±0.1 W m$^{-2}$, and is applied continuously over the
surface area of the Earth (5.10×10$^{14}$ m$^2$). For comparison, the heat gain obtained in IPCC AR5
amounts to $274 \pm 78$ ZJ and 0.4 W m$^{-2}$ over the period 1971–2010 (Rhein et al., 2013). In IPCC
AR6, the total heat rate has been assessed by 0.57 [0.43 to 0.72] W m$^{-2}$ for the period 1971-2018
(Forster et al., 2021). We further infer a total heating rate of $0.76 \pm 0.2$ W m$^{-2}$ for the most recent
era 2006-2020.
Thus, the number of how fast heat has been accumulated in the Earth system has increased during
the most recent era as compared to the long-term estimate – an outcome which reconfirms the
earlier finding in von Schuckmann et al. (2020), and which had then been concurrently and
independently confirmed in Foster et al. (2021), Hakuba et al. (2021), Loeb et al. (2021) and
Kramer et al. (2021). The drivers of a larger EEI in the 2000s than in the long-term period since
1971 are still unclear, and several mechanisms are discussed in literature. For example, Loeb et al.
(2021) argue for a decreased reflection of energy back into space by clouds and sea-ice, and
increases in well-mixed greenhouse gases (GHG) and water vapor to account for this increase in
EEI. (Kramer et al., 2021) refers to a combination of rising concentrations of well-mixed GHG
and recent reductions in aerosol emissions accounting for the increase, and (Liu et al., 2020)
addresses changes in surface heat flux together with planetary heat re-distribution and changes in
ocean heat storage. Future studies are needed to further explain the drivers of this change, together
with its implications for changes in the Earth system.



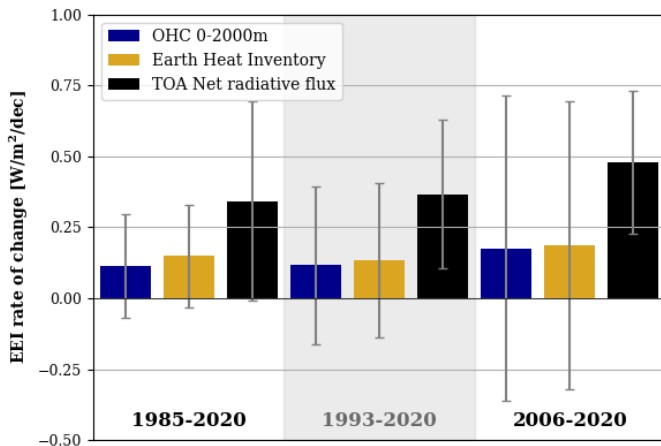

*Figure 9: Decadal scale rate of change for the Earth Energy Imbalance (EEI) in Wm⁻²/decade as*
*derived from the Earth heat inventory in Fig. 8 (yellow), OHC of the 0-2000m depth layer (blue,*
*see section 2) and net flux at the top of the atmosphere (TOA, black) based on the estimates of Liu*
*et al. (2020) and Loeb et al. (2021) for three different periods, 1985-2020 (i.e., the full available*
*net flux at TOA estimate), 1993-2020 (i.e., the altimeter era) and 2006-2020 (i.e., the GCOS golden*
*period for the Earth heat inventory). See Liu et al. (2020) and Loeb et al. (2021) on more details*
*of the uncertainty estimate, and note that satellite instrument drift error is not considered. A linear*
*regression has been applied to obtain the rate of change.*

With respect to the current status of the GCOS, we further want to emphasize the fact that today
the Earth heat inventory is the best estimate for the absolute value of the Earth energy imbalance.
This is explained by the fact that satellite derived measurements for the net flux at the top of the
atmosphere (TOA) have to be anchored by an absolute value, which is done through the use of the
Earth heat inventory, and for which mostly global OHC is used (Loeb et al., 2012; 2021; Liu et
al., 2020). However, the temporal change of the EEI can be best estimated from the net flux at
TOA from remote sensing data as these are superior in terms of temporal stability. To further
discuss the temporal change of the EEI, we compare our results of the Earth heat inventory with
the satellite derived net flux at TOA. Consistent with the results of Loeb et al. (2021), the net flux
estimates at TOA show a change in the EEI at a significant rate of $0.48 \pm 0.3$ Wm⁻²/decade during
the period 2006-2020. In 1985-2020 (1993-2020), the value amounts to $0.3 \pm 0.4$ Wm⁻²/decade
($0.4 \pm 0.3$ Wm⁻²/decade) (Fig. 9).

For the Earth heat inventory, the results show that uncertainties for estimating temporal changes
of the EEI are still too large to obtain significant results, even during the GCOS 'golden period'
for the Earth heat inventory in 2006-2020 (Loeb et al., 2022). In other words, this comparison
highlights the strength of the complementary use of different independent GCOS components. But
the results also show that the current status of the GCOS does not allow for unraveling the rate of
change of heat stored in the Earth system components, which is critical information to further
understand associated changes in the Earth system (Fig. 1), and to validate climate models for



improving projections of these changes into the future. Hence, these results further underpin the
need for sustaining and further extending the GCOS for improving our knowledge and monitoring
capacity of estimates for how much and where heat is stored in the Earth system.
Besides heat, which is the focus of this study, Earth also stores energy chemically through
photosynthesis in living and dead biomass with plant growth. Recent studies (Crisp et al., 2022;
Denning, 2022; Friedlingstein et al., 2022) on the Global Carbon Budget and cycle show that
approximately 25% of the added anthropogenic CO2 is removed from the atmosphere by increased
plant growth, which is a result of fertilization by rising atmospheric CO2 and Nitrogen inputs and
of higher temperatures and longer growing seasons in northern temperate and boreal areas
(Friedlingstein et al., 2022). This significant increase in carbon uptake by the biosphere indicates
that more energy is stored inside biomass, together with the stored carbon. The quantification of
the additional amount of energy stored inside the biosphere is outside the scope of this study.
**7. Data availability**
The time series of the Earth heat inventory are published at DKRZ (https://www.dkrz.de/, last
access: 20 July 2020) under https://www.wdc-climate.de/ui/entry?acronym=GCOS_EHI_1960-
2020, more precisely for:
• (von Schuckmann et al., 2022); data for ocean heat content (section 2), and the total heat
inventory as presented in section 6 are integrated.
• (Kirchengast et al., 2022) ; data for the atmospheric heat content are distributed (section
3).
• (Cuesta Valero et al., 2022c); data for the ground heat storage, together with the total
continental heat gain are provided (section 4)
• (Vanderkelen et al., 2022); data for inland freshwater heat storage is included (section 4)
• (Nitzbon et al., 2022b); data for permafrost are delivered (section 4).
• (Adusumilli et al., 2022); data for the cryosphere heat inventory are provided.
Persistent identifiers (PIDs) for the specific data access are provided in Table 2.

| Earth heat inventory component | PID | Reference |
|---|---|---|
| **Ocean heat content; Total Earth heat inventory** | https://hdl.handle.net/21.14106/9b2fddbe4637e3bb9fbf2414c55e6aad0e3923b0 | von Schuckmann et al., 2022 |
| **Atmospheric heat content** | https://hdl.handle.net/21.14106/2c4e7216177fcb742f324eae2792c43faf8361f1 | Kirchengast et al., 2022 |
| **Continental heat content** | https://hdl.handle.net/21.14106/302a4aedadcabf09d5f432003361275e9102a48a | Cuesta Valero et al., 2022c |
| **Inland water heat content** | https://hdl.handle.net/21.14106/e095f83398baa6e5b355ba88ae97cd7dedd008de | Vanderkelen et al., 2022 |
| **Heat available to melt permafrost** | https://hdl.handle.net/21.14106/a9654c3d10c0002da4dde3ef080f6503e2deebf5 | Nitzbon et al., 2022b |
| **Heat available to melt the cryosphere** | https://hdl.handle.net/21.14106/b9829ba3230f0631d3545a66a88e1c89803510ee | Adusumilli et al., 2022 |




*Table 2: Overview on persistent identifiers (PIDs) for data access for each component of the Earth heat inventory. The results are presented in Fig. 8.*

## 8. Conclusion

This study builds on the first internationally and multidisciplinary driven Earth heat inventory in 2020 (von Schuckmann et al., 2020) and provides an update on total Earth system heat accumulation, heat storage in all Earth system components (ocean, land, cryosphere, atmosphere) and the Earth energy imbalance up to the year 2020. Moreover, this study succeeded to improve estimates, to further extent and foster international collaboration, and to continue to move towards a more complete view on where and how much heat is stored in the Earth system through the addition of new estimates such as for permafrost thawing, inland freshwater (section 4) and Antarctic sea ice (section 5). Results obtained reveal a total Earth system heat gain of 380±62 ZJ over the period 1971–2020, with an associated total heating rate of 0.48±0.1 W m$^{-2}$. 89 ± 17 % of this heat stored in the ocean, 6 ± 0.1 % on land, 4 ± 1% in the cryosphere and 1 ± 0.2 % in the atmosphere (Fig. 8, 11). The analysis additionally reconfirms an increased heating rate which amounts to 0.76 ± 0.2 W/m$^{-2}$ for the most recent era 2006-2020. These results are consistent with previous estimates, which is again demonstrated through a comprehensive assessment of estimates for the EEI published in peer-reviewed literature (Fig. 10). Albeit the drivers for this change still need to be elucidated and most likely reflect the interplay between natural variability and anthropogenic change (Loeb et al., 2021; Kramer et al., 2021; Liu et al., 2020), their implications for changes in the Earth system are reflected in the many record levels of change in the 2000s reported elsewhere, e.g., (Cheng et al., 2022; Forster et al., 2022; Gulev et al., 2021; WMO, 2022).

The recent Glasgow Climate Pact (UNFCCC, 2021) is '*Acknowledging that climate change is a common concern of humankind …*' and '*Recognizing … the importance of international cooperation in addressing climate change and its impacts…*', and '*Recognizes the importance of the best available science for effective climate action and policy making*'. The UN 2030 Agenda for Sustainable Development[11] states that climate change is "*one of the greatest challenges of our time ...*" and warns *"... the survival of many societies, and of the biological support systems of the planet, is at risk*". The outcome document of the Rio+20 Conference, The Future We Want[12], defines climate change as "*an inevitable and urgent global challenge with long-term implications for the sustainable development of all countries*". The Paris Agreement builds upon the United Nations Framework Convention on Climate Change and for the first time all nations agreed to undertake ambitious efforts to combat climate change, with the central aim to keep global temperature rise this century well below 2 °C above pre industrial levels and to limit the temperature increase even further to 1.5 °C. Article 14 of the Paris Agreement requires the Conference of the Parties serving as the meeting of the Parties to the Paris Agreement (CMA) to periodically take stock of the implementation of the Paris Agreement and to assess collective

---

[11]

https://sustainabledevelopment.un.org/content/documents/21252030%20Agenda%20for%20Sustainable%20Development%20web.pdf

[12]     https://sustainabledevelopment.un.org/content/documents/733FutureWeWant.pdf

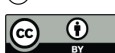

progress towards achieving the purpose of the agreement and its long-term goals through the so-
called Global Stocktake of the Paris Agreement (GST)[13] based on best available science.
The Earth heat inventory provides information on how much and where heat is accumulated and
stored in the Earth system. Moreover, it provides a measure of how much the Earth is out of energy
balance, and when combined with directly measured net flux at the top of the atmosphere, enables
also to understand the change of the EEI over time. This in turn allows for assessing the portion of
the anthropogenic forcing that the Earth's climate system has not yet responded to (Hansen et al.,
2005) and defines additional global warming that will occur without further change in human-
induced forcing (Hansen et al., 2017). The Earth heat inventory is thus one of the key critical global
climate change indicators defining the prospects for continued global warming and climate change
(Hansen et al., 2011; von Schuckmann et al., 2016; 2020) Hence, we call for an implementation
of the Earth heat inventory into the global stocktake.

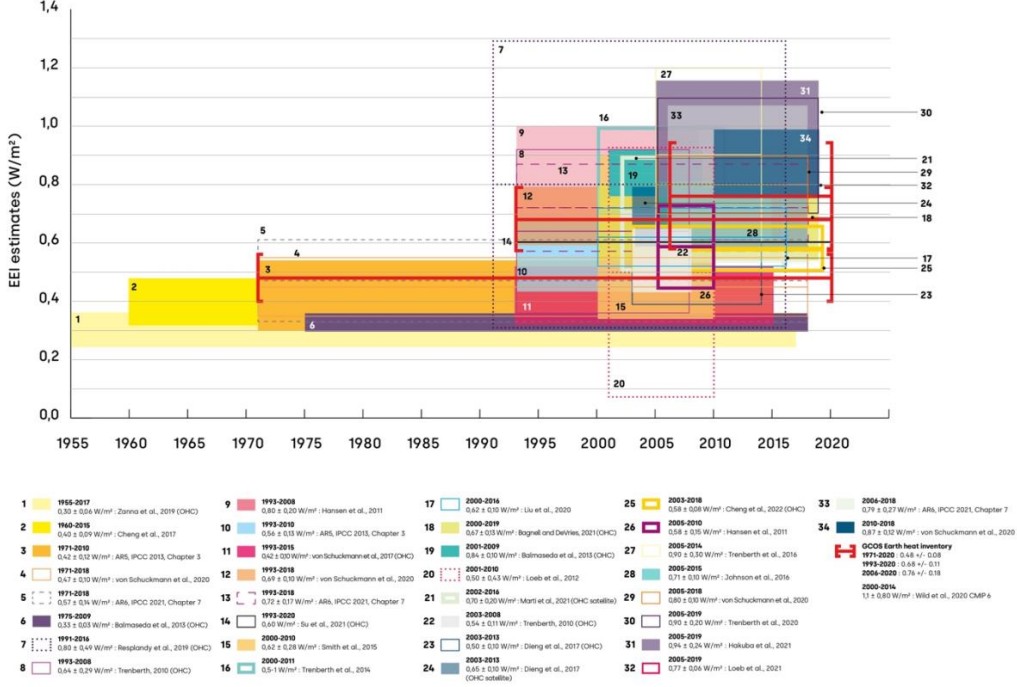

**Figure 10:** *Overview on EEI and d(OHC)/dt (indicated with (OHC) in the legend) estimates as*
*obtained from previous publications; references are listed in the figure legend. The color bars take*
*into account the uncertainty ranges provided in each publication, respectively. For comparison,*

---

[13]    https://unfccc.int/topics/global-stocktake/global-
stocktake#:~:text=The%20global%20stocktake%20of%20the,term%20goals%20(Article%2014)
.



*the estimates of our Earth heat inventory based on the results of Fig. 8 have been added (red lines)*
*for the periods 1971–2020, 1993–2020 and 2006–2020.*



***Figure 11:*** *Schematic presentation on the Earth heat inventory for the current anthropogenically*
*driven positive Earth energy imbalance at the top of the atmosphere (TOA). The relative partition*
*(in %) of the Earth heat inventory presented in Fig. 8 for the different components is given for the*
*ocean (upper: 0–700 m, intermediate: 700–2000 m, deep: >2000 m), land, cryosphere (grounded*
*and floating ice) and atmosphere, for the periods 2006–2020 and 1971–2020 (for the latter period*
*values are provided in parentheses), as well as for the EEI. The total heat gain (in red) over the*
*period 1971–2020 is obtained from the Earth heat inventory as presented in Fig. 8.*
The quantifications presented in this study are the result of multidisciplinary global-scale
collaboration and demonstrate the critical importance of concerted international efforts for climate
change monitoring and community-based recommendations for the GCOS. For the GOOS, the
core Argo sampling needs to be sustained – which includes the maintenance of shipboard
collection of reference data for validation - and complemented by remote sensing data. Extensions



such as into the deep ocean layer need to be further fostered, and technical developments for the
measurements under ice and in shallower areas need to be sustained and extended. Moreover,
continued efforts are needed to further advance bias correction methodologies, uncertainty
evaluations, data recovery and processing of the historical dataset.
For the ground heat storage, the estimate had been hampered by a lack of subsurface temperature
profiles in the southern hemisphere, as well as by the fact that most of the profiles were measured
before the 2000s. Subsurface temperature data are direct and independent (not proxy)
measurements of temperature yielding information on the temporal variation of the ground surface
temperature and ground heat flux at the land surface. A larger spatial scale dataset of the thermal
state of the subsurface from the last millennium to the present will aid in the continuing monitoring
of continental heat storage, provide initial conditions for Land Surface Model (LSM) components
of Earth System Models (ESMs) (Cuesta-Valero et al., 2019), and serve as a dataset for validation
of climate models' simulations (Cuesta-Valero et al., 2021; Cuesta-Valero et al., 2016). Progress
in understanding climate variability through the last millennium must lean on additional data
acquisition as the only way to reduce uncertainty in the paleoclimatic record and on changes to the
current state of the continental energy reservoir. Remote sensing data are expected to be very
valuable to retrieve recent past and future changes in ground heat flux at short-time scales with
near global coverage. However, collecting subsurface temperature data is urgent as we must make
a record of the present thermal state of the subsurface before the subsurface climate baseline is
affected by the downward propagating thermal signal from current climate heating. Furthermore,
an international organization should take responsibility to gather and curate all measured
subsurface temperature profiles currently available and those that will be measured in the future,
as the current practices, in which individual researchers are responsible for measuring, storing and
distributing the data, have led to fragmented datasets, restrictions in the use of data, and loss of the
original datasets. Support from GCOS for an international data acquisition and curating efforts
would be extremely important in this context.
For the permafrost estimates, the primary sources of uncertainty arise from lacking information
about the amount and distribution of ground ice in permafrost regions, as well as measurements of
liquid water content (Nitzbon et al., 2022). Permafrost heat storage is defined as the required heat
to change the mass of ground ice at a certain location, thus monitoring changes in ground ice and
water contents would be required to improve estimates of this component of the continental heat
storage. Nevertheless, the current monitoring system for permafrost soils is focused on soil
temperature, and the distribution of stations is still relatively scarce in comparison with the vast
areas that need to be surveyed (Biskaborn et al., 2015). Due to the current limitations in the
observational data, a permafrost model was used to estimate the heat uptake by thawing of ground
ice. This approach retrieves latent heat fluxes in extensive areas and at depths relevant to analyze
the long-term change in ground ice mass, but at the cost of ignoring other relevant processes, such
as ground subsidence, to balance model performance with computational resources. Including
permafrost heat storage in the Tibetan Plateau is a priority for the next iteration of this work, as
well as to explore new methods to evaluate model simulations using the available observations in
permafrost areas.
For inland water heat storage, a better representation of lake and reservoir volume would be
possible by better accounting for lake bathymetry using the GLOBathy (Khazaei et al., 2022)



dataset and results from the upcoming Surface Water and Ocean Topography (SWOT) mission.
These improvements in the representation of lake volume, and an updated lake mask will be
available in the upcoming ISIMIP3 simulation round, next to improved meteorological forcing
data (Golub et al., 2022). In contrast to (Vanderkelen et al., 2020), the heat storage in rivers is not
included in this analysis due to the high uncertainties in simulated river water volume. To reduce
the uncertainty in river heat storage, the estimation of river water storage should be improved,
together with an explicit representation of water temperature in the global hydrological models
(Wanders et al., 2019).  These improvements will be incorporated in ISIMIP3 and will lead to
better estimates of inland water heat storage, thus enhancing future estimates of continental heat
storage. In the long run, these model-based estimates could be supplemented or replaced by
observation-based estimates, which would however require a large, global-scale effort to monitor
lake and river temperatures at high spatial resolution and over long time periods.
For the cryosphere, sustained remote sensing for all of the cryosphere components is critical in
quantifying future changes over these vast and inaccessible regions; in situ observations are also
needed for process understanding and in order to properly calibrate and validate them. For sea ice,
observations of the albedo, the area and ice thickness are all essential - the continuation of satellite
altimeter missions with high inclination, polar focused orbits is critical in our ability to monitor
sea ice thickness in particular. Observations of snow thickness with multi-frequency altimeters are
essential for further constraining sea ice thickness estimates. For ice sheets and glaciers, reliable
gravimetric, geodetic, and ice velocity measurements, knowledge of ice thickness and extent,
snow/firn thickness and density, and the continuation of the now three-decade long satellite
altimeter record are essential in understanding changes in the mass balance of grounded and
floating ice. The recent failure of Sentinel-1b, which in tandem with Sentinel-1a could be used to
systematically measure ice speed changes every 6 days, means that images are now being acquired
every 12 days and thus an earlier launch of Sentinel-1c should be encouraged to regain the ability
to monitor ice speed changes over short time-scales. The estimate of glacier heat uptake is
particularly affected by lacking knowledge of ice melt below sea level, and to a lesser degree,
lacking knowledge of firn and ice temperatures. This lack of observations is likely related to most
studies on glaciers focussing on their contribution to sea-level rise or seasonal water availability,
where melt below sea level and warming of ice do not matter much. However, it becomes obvious
here that this gap introduces a systematic bias in the estimate of cryospheric energy uptake, which
is presumably small compared to the other components, but unconstrained. Although the Antarctic
sea ice change and the warming of Greenland and Antarctic firn are poorly constrained or have
not significantly contributed to this assessment, they may become increasingly important over the
coming decades. Similarly, there exists the possibility for rapid change associated with positive
ice dynamical feedbacks at the marine margins of the Antarctic Ice Sheet. Sustained monitoring
of each of these components will, therefore, serve the dual purpose of furthering the understanding
of the dynamics and quantifying the contribution to Earth's energy budget. In addition to data
collection, open access to the data and data synthesis products, as well as coordinated international
efforts, are key to the continued monitoring of the ice loss from the cryosphere and its related
energy uptake.
For the atmosphere, there is a need to sustain and enhance a coherent operational long-term
monitoring system for the provision of climate data records of essential climate variables.
Observations from radiosonde stations within the GCOS reference upper air network (GRUAN)



and from satellite-based GNSS radio occultation deliver thermodynamic profiling observations of
benchmark quality and stability from surface to stratopause. For climate monitoring, it is of critical
importance to ensure continuity of such observations with global coverage over all local times.
This continuity of radio occultation observations in the future is not sufficiently guaranteed as we
are facing an imminent observational gap in mid- to high latitudes for most local times(IROWG,
2021), which is a major concern. Thus, there is an urgent need for satellite missions in high
inclination orbits to provide full global and local time coverage in order to ensure global climate
monitoring. Operational radio occultation missions need to be maintained as backbone for a global
climate observing system and long-term availability and archiving of measurement data, metadata
and processing information needs to be ensured.
In summary, we also call for urgently needed actions for enabling continuity, archiving, rescuing
and calibrating efforts to assure improved and long-term monitoring capacity of the GCOS for the
Earth heat inventory. Particularly, the summarized recommendations include
• Need to sustain, reinforce or even to establish data repositories for historical climate data
(archiving)
• Need to reinforce efforts for recovery projects for historical data and associated meta-data
information (rescuing)
• Need to sustain and reinforce the GCOS for assuring the monitoring of the Earth heat
inventory targets (continuity)
• Need to foster calibration measurements (in situ) for assuring quality and reliability of
large-scale measurement techniques (e.g., remote sensing, autonomous components (eg
argo) (calibrating)
A continuous effort to regularly update the Earth heat inventory is important as this global climate
indicator crosses multidisciplinary boundaries and calls for the inclusion of new science
knowledge from the different disciplines involved, including the evolution of climate observing
systems and associated data products, uncertainty evaluations, and processing tools. The outcomes
have further demonstrated how we are able to evolve our estimates for the Earth heat inventory
while bringing together different expertise and major climate science advancements through a
concerted international effort. All of these component estimates are at the leading edge of climate
science. Their union has provided a new and unique insight on the inventory of heat in the Earth
system, its evolution over time and the absolute values. The data product of this effort is made
available and can be thus used for model validation purposes.
This study has demonstrated the unique value of such a concerted international effort, and we thus
call for a regular evaluation of the Earth heat inventory. This updated attempt presented here has
been focused on the global area average only, and evolving into regional heat storage and
redistribution, the inclusion of various timescales (e.g., seasonal, year to year) and other climate
study tools (e.g., indirect methods, ocean reanalyses) would be an important asset of this much
needed regular international framework for the Earth heat inventory. This would also respond
directly to the request of GCOS to establish the observational requirements needed to further
monitor the Earth's cycles and the global energy budget (GCOS, 2021). The outcome of this study
will therefore directly feed into GCOS' assessments of the status of the global climate observing
system, and the identified observation requirements will guide the development of the next



generation of in situ and satellite global climate observations as specified by GCOS by all national
meteorological services and space agencies and other oceanic and terrestrial networks.

**Acknowledgements.**

Ocean: OHC estimate from the product ISAS (Gaillard et al., 2016) was provided by 'Service National d'Observation Argo France' (INSU/CNRS) at OSU IUEM (https://www.argo-france.fr/).

Atmosphere: We acknowledge the WEGC EOPAC team for providing the OPSv5.6 RO data (available online at https://doi.org/10.25364/WEGC/OPS5.6:2020.1) as well as quality-processed Vaisala RS data, UCAR/CDAAC (Boulder, CO, USA) for access to RO phase and orbit data, ECMWF (Reading, UK) for access to operational analysis and forecast data, ERA5 reanalysis data, and RS data from the ERA-Interim archive, JMA (Tokyo, Japan) for provision of the JRA55 and JRA55C reanalysis data, and NASA GMAO (Greenbelt, MD, USA) for access of the MERRA-2 reanalysis data.

**Financial support.**

Maximilian Gorfer was supported by WEGC atmospheric remote sensing and climate system research group young scientist funds. Michael Mayer was supported by Austrian Science Fund project P33177.

Donata Giglio and Mikael Kuusela acknowledge support from NOAA (Award NA21OAR4310261).

L.C. acknowledges financial supports from the Strategic Priority Research Program of the Chinese Academy of Sciences (XDB42040402), National Natural Science Foundation of China (grant number 42122046, 42076202).

J.C. and Y.L. were supported by the Centre for Southern Hemisphere Oceans Research (CSHOR), jointly funded by the Qingdao National Laboratory for Marine Science and Technology (QNLM, China) and the Commonwealth Scientific and Industrial Research Organisation (CSIRO, Australia), and the Australian Research Council's Discovery Project funding scheme (project DP190101173). TMcD and PMB gratefully acknowledge Australian Research Council support through grant FL150100090. This paper contributes to the tasks of the Joint SCOR/IAPSO/IAPWS Committee on the Thermophysical Properties of Seawater.

Hugo Beltrami was supported by grants from the National Sciences and Engineering Research Council of Canada Discovery Grant (NSERC DG 140576948) and the Canada Research Chairs Program (CRC 230687). Hugo Beltrami holds a Canada Research Chair in Climate Dynamics

Francisco José Cuesta-Valero is an Alexander von Humboldt Research Fellow at the Helmholtz Centre for Environmental Research (UFZ).

Richard P. Allan is funded by the National Centre for Earth Observation RCUK grant NE/RO16518/1.

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

von Schuckmann, Karina; Minière, Audrey; Gues, Flora; Cuesta-Valero, Francisco; Kirchengast,
Gottfried; Adusumilli, Susheel; Straneo, Fiammetta; Allan, Richard; Barker, Paul M.;
Beltrami, Hugo; Boyer, Tim; Cheng, Lijing; Church, John; Desbruyeres, Damien; Dolman,
Han; Domingues, Catia; García-García, Almudena; Gilson, John; Gorfer, Maximilian;
Haimberger, Leopold; Hendricks, Stefan; Hosoda, Shigeki; Johnson, Gregory; Killick,
Rachel; King, Brian; Kolodziejczyk, Nicolas; Korosov, Anton; Krinner, Gerhard; Kuusela,
Mikael; Langer, Moritz; Lavergne, Thomas; Lawrence, Isobel; Li, Yuehua; Lyman, John;





Marzeion, Ben; Mayer, Michael; MacDougall, Andrew; McDougall, Trevor; Monselesan,
Didier; Nitzbon, Jean; Otosaka, Inès; Peng, Jian; Purkey, Sarah; Roemmich, Dean; Sato,
Kanako; Sato, Katsunari; Savita, Abhishek; Schweiger, Axel; Shepherd, Andrew;
Seneviratne, Sonia; Slater, Donald; Slater, Thomas; Smith, Noah; Steiner, Andrea; Szekely,
Tanguy; Suga, Toshio; Thiery, Wim; Timmermanns, Mary-Louise; Vanderkelen, Inne;
Wijffels, Susan; Wu, Tonghua; Zemp, Michael; Simons, Leon (2022). Heat stored in the
Earth system 1960-2020: Where does the energy go?. World Data Center for Climate
(WDCC) at DKRZ. https://www.wdc-climate.de/ui/entry?acronym=GCOS_EHI_1960-
2020_OHC
von Schuckmann, K., Palmer, M. D., Trenberth, K. E., Cazenave, A., Chambers, D.,
Champollion, N., Hansen, J., Josey, S. A., Loeb, N., Mathieu, P.-P., Meyssignac, B., &
Wild, M. (2016). An imperative to monitor Earth's energy imbalance. *Nature Climate*
*Change*, *6*(2), 138–144. https://doi.org/10.1038/nclimate2876
von Schuckmann, K, Cheng, L., Palmer, M. D., Hansen, J., Tassone, C., Aich, V., Adusumilli,
S., Beltrami, H., Boyer, T., Cuesta-Valero, F. J., Desbruyères, D., Domingues, C., García-
García, A., Gentine, P., Gilson, J., Gorfer, M., Haimberger, L., Ishii, M., Johnson, G. C., …
Wijffels, S. E. (2020). Heat stored in the Earth system: where does the energy go? *Earth*
*Syst. Sci. Data*, *12*(3), 2013–2041. https://doi.org/10.5194/essd-12-2013-2020
von Schuckmann, K, & Le Traon, P.-Y. (2011). How well can we derive Global Ocean
Indicators from Argo data? *Ocean Sci.*, *7*(6), 783–791. https://doi.org/10.5194/os-7-783-
2011
von Schuckmann, Karina, Le Traon, P.-Y., Smith, N., Pascual, A., Brasseur, P., Fennel, K.,
Djavidnia, S., Aaboe, S., Fanjul, E. A., Autret, E., Axell, L., Aznar, R., Benincasa, M.,
Bentamy, A., Boberg, F., Bourdallé-Badie, R., Nardelli, B. B., Brando, V. E., Bricaud, C.,
… Zuo, H. (2018). Copernicus Marine Service Ocean State Report. *Journal of Operational*
*Oceanography*, *11*(sup1), S1–S142. https://doi.org/10.1080/1755876X.2018.1489208
Wanders, N., Thober, S., Kumar, R., Pan, M., Sheffield, J., Samaniego, L., & Wood, E. F.
(2019). Development and Evaluation of a Pan-European Multimodel Seasonal Hydrological
Forecasting System. *Journal of Hydrometeorology*, *20*(1), 99–115.
https://doi.org/10.1175/JHM-D-18-0040.1
Wang, X., Key, J., Kwok, R., & Zhang, J. (2016). Comparison of Arctic Sea Ice Thickness from
Satellites, Aircraft, and PIOMAS Data. In *Remote Sensing* (Vol. 8, Issue 9).
https://doi.org/10.3390/rs8090713
WCRP Global Sea Level Budget Group. (2018). Global sea-level budget 1993–present. *Earth*
*Syst. Sci. Data*, *10*(3), 1551–1590. https://doi.org/10.5194/essd-10-1551-2018
Webster, M. A., DuVivier, A. K., Holland, M. M., & Bailey, D. A. (2021). Snow on Arctic Sea
Ice in a Warming Climate as Simulated in CESM. *Journal of Geophysical Research:*
*Oceans*, *126*(1), e2020JC016308. https://doi.org/https://doi.org/10.1029/2020JC016308
Weihs, P., Laimighofer, J., Formayer, H., & Olefs, M. (2021). Influence of snow making on
albedo and local radiative forcing in an alpine area. *Atmospheric Research*, *255*, 105448.
https://doi.org/https://doi.org/10.1016/j.atmosres.2020.105448
WGMS. (2021). *Fluctuations of Glaciers Database. World Glacier Monitoring Service, Zurich,*
*Switzerland.* https://doi.org/DOI:10.5904/wgms-fog-2021-05
Wijffels, S., Roemmich, D., Monselesan, D., Church, J., & Gilson, J. (2016). Ocean temperatures
chronicle the ongoing warming of Earth. *Nature Climate Change*, *6*(2), 116–118.
https://doi.org/10.1038/nclimate2924



Willis, J. K., Roemmich, D., & Cornuelle, B. (2004). Interannual variability in upper ocean heat
content, temperature, and thermosteric expansion on global scales. *Journal of Geophysical*
*Research: Oceans*, *109*(C12). https://doi.org/10.1029/2003JC002260
Wilson, N., Straneo, F., & Heimbach, P. (2017). Satellite-derived submarine melt rates and mass
balance (2011–2015) for Greenland's largest remaining ice tongues. *The Cryosphere*, *11*,
2773–2782. https://doi.org/10.5194/tc-11-2773-2017
WMO. (2022). *The State of the Global Climate 2021*.
https://library.wmo.int/index.php?lvl=notice_display&id=22080
Wunsch, C. (2020). Is the Ocean Speeding Up? Ocean Surface Energy Trends. *Journal of*
*Physical Oceanography*, *50*, 3205–3217. https://doi.org/10.1175/JPO-D-20-0082.1
Zanna, L., Khatiwala, S., Gregory, J. M., Ison, J., & Heimbach, P. (2019). Global reconstruction
of historical ocean heat storage and transport. *Proceedings of the National Academy of*
*Sciences*, *116*(4), 1126. https://doi.org/10.1073/pnas.1808838115
Zemp, M., Huss, M., Thibert, E., Eckert, N., McNabb, R., Huber, J., Barandun, M., Machguth,
H., Nussbaumer, S. U., Gärtner-Roer, I., Thomson, L., Paul, F., Maussion, F., Kutuzov, S.,
& Cogley, J. G. (2019). *Global and regional glacier mass changes from 1961 to 2016*.
https://doi.org/10.5281/ZENODO.3557199
Zemp, Michael, Huss, M., Eckert, N., Thibert, E., Paul, F., Nussbaumer, U. S., & Gärtner-Roer,
I. (2020). Brief communication: Ad hoc estimation of glacier contributions to sea-level rise
from the latest glaciological observations (I5946, trans.). *Cryosphere*, *14*(3).
https://doi.org/10.5194/tc-14-1043-2020
Zhang, J., & Rothrock, D. A. (2003). Modeling Global Sea Ice with a Thickness and Enthalpy
Distribution Model in Generalized Curvilinear Coordinates. *Monthly Weather Review*,
*131*(5), 845–861. https://doi.org/10.1175/1520-0493(2003)131<0845:MGSIWA>2.0.CO;2
Zhang, R., Wang, H., Fu, Q., Rasch, J. P., & Wang, X. (2019). Unraveling driving forces
explaining significant reduction in satellite-inferred Arctic surface albedo since the 1980s.
*Proceedings of the National Academy of Sciences*, *116*(48), 23947–23953.
https://doi.org/10.1073/pnas.1915258116
Zou, C.-Z., Xu, H., Hao, X., & Fu, Q. (2021). Post-Millennium Atmospheric Temperature
Trends Observed From Satellites in Stable Orbits. *Geophysical Research Letters*, *48*(13),
e2021GL093291. https://doi.org/https://doi.org/10.1029/2021GL093291