# Peer review of "Heat stored in the Earth system 1960-2020: Where does the energy go?"

_Earth System Science Data, 2022_

## Author Comment (AC1)

Point-by-point reply reviewer 1

Very positive about this product! Remarkable compilation addressing a most-urgent topic. I will definitely recommend for publication in ESSD but I need to recommend changes that, in my estimation, strengthen and clarify the manuscript for many readers. Apologies for long list. First, I address cross-cutting issues. Later, I address technical changes in each section.

We would like to gratefully thank the reviewer for the comments, and the time allocated to provide specific recommendations to help improving the manuscript, and below we provide a point-by-point reply to each of the comments.

1.) Manuscript seems way too long. 1330 lines from start of title to end of acknowledgements/ financials in this version, compared to 1150 in previous version (von Schuckmann et al. 2020). Nearly 16% increase? Not good for authors nor for ESSD. We might expect a percent or two, from additional relevant materials, but offset substantially by the fact that authors will not need to repeat here what they already described (in detail) in previous version. Many specific comments related to length follow but - overall - possible to easily reduce by > 20%? Readers do not want or need to wander through so much material. We want crisp sharp accurate restatement and update of EEI! Tricky balancing act: you want this one to stand alone, conveying accurate info for any new reader, but you also want to not repeat here background material from v1. See several shortening suggestions under technical comments.

Thank you very much for this recommendation. Indeed, we need to improve the balancing act between a 'stand-alone' manuscript, and the objective for regular update on the Earth heat inventory. A major reason for the increase in length as compared to the first publication is the strengthened collaboration (we have increased the list of expertise), and the new additions we did not cover in the previous analysis (e.g., permafrost, inland water storage). However, we agree with the reviewer's point made on this aspect, and we will follow suggestions by the reviewer as replied also below. We hope that we could with this revision sharpen the messaging, and manuscript.

2.) For this reader, way too much GCOS advocacy! I write as a big fan of GCOS, recognizing that WMO dismissed or chased away good people. I endorse every concern about GCOS, but these concerns do not belong in ESSD! One strategy: mention (one sentence) relevant ECVs in each land, ocean, ice, atmosphere section, then summarize concerns in sentence or two of conclusions/recommendations? At a minimum, and assuming time is short, end with a clear simple sentence of concern?

Thank you for the comment. We will sharpen the message for the framework aspect, but we also want to highlight that this activity is endorsed under GCOS, which is a critical aspect of maintaining and soliciting (non-funded) international expertise on this one specific topic. We hope that with the proposed changes we could balance our needs, and adequately replies to the reviewers comment.

3.) Likewise, for this reader, way too much IPCC content? Readers will know (will have participated in) AR5 or AR6 or both, plus SR. EEI stands on its own! IPCC will cite it, not the other way around. Use IPCC citations sparingly, remove all IPCC quotes. These do not contribute to crisp sharp update of EEI.

The IPCC quotes have been now removed.

    4.)  Settle on standard common definitions of time periods.

We would like to thank the reviewer for having spotted this issue, and please find below specific replies, and at the end a synthesis of further revisions to meet this comment.

- Confusion begins in abstract (line 90: 2006-2020 does not constitute a decade),

Reworded to 'most recent period'.

- continues around line 157 (here you bounce between 1971-2018 and 2010-2018),

Not relevant anymore, removed from introduction now.

- emerges again at line 315 (era of Argo floats starts in 2000 but references not until 2016 or 2019?),

Comment not clear, can you please precise ?

- again at line 340 (near-global coverage of Argo floats by 2005 but you already said this slightly differently at lines 318, 319),

Yes, this is important, w state both, 2005 and 2006 – this is to explain why we start in 2006 – best available observing system. Precisely explained later in the text, and in the figure caption.

- at line 391 (legend to Fig 3 where you define 1960-2020 as historical, 1993-2020 as altimeter, and 2006-2020 as recent [= Argo?],

Yes, this are the periods we use for the heat content estimate, and the legend clarifies that 2006 is the golden Argo era.

- likewise and expanded at line 412),
- at line 421 (25 years for historical [!!??] and altimetric and 15 years [not = decadal!] for Argo), at line 453 (Table 1, in which 1971-2020 emerges again),

The 25 year are linked to the bandwidth of the LOWESS approach, not the actual time period.

- at lines 482 and 490 (introduces 1990 as boundary for deep OHC except at line 495 you use 1993),

Thank you, and we have corrected this 1990.

- at line 512 atmosphere (now using 1980 to 2020),

Note that this is not yet a discussion of the atmospheric datasets used; it is a sentence as part of the introductory paragraphs of the atmosphere section 3 that reports on other published results which refer to this period.

- line 561 (still AHC, now back-extended 1980 to 1960),

This first paragraph to the dataset subsection 3.2 explains, as an overall intro relative to the previous von Schuckmann et al. (2020) paper, that this backward extension is needed to now achieve the improved 1960 onwards estimate, and that we now have 1960 to 2020 data.

- line 574 (talking about coverage periods of reanalysis products including specifically ERA-5), line 582 (RO and RS products 2002-2020 and 1996-2000 respectively),

Yes, these paragraphs concisely introduce the individual datasets and their timespans; in one sentence each. We consider it simplest this way (though would contribute to an overall table in a next updated energy inventory paper version), to have these dataset timespans of subsection 3.2 clearly separate from the important selected time periods that we use in the following AHC results subsection 3.3. In this results subsection, we clearly state the chosen analysis & results time periods at the very beginning, to have this clear and used it consistently over Figure 4, Figure 5 and Table 2, throughout the section (see answers to the next three comments).

- line 618 (Fig 4, all panels using 1960-2020),

See comment on line 561 above and please note that the entry sentence to this subsection 3.3 "Atmospheric heat content change since 1960 and its amplification" clearly states that the inventory shown in Figure 4 spans in total over 1960 to 2020; plus that it shows "trends for selected periods" that "align with those for ocean data" (Fig. 3 data) and available atmospheric datasets. These selected time periods used in Fig. 4 (see panel legends and trendlines shown) were then consistently used as well for the other AHC-related results highlighted in Fig. 5 and summarized in Table 2.

- line 645 (Fig 5, now using time periods 1993-2020, 2001-2020 and 2006-2020 without categorization but as anomalies from 1961-2000?),

This evaluation has been applied with respect to the atmospheric analysis for specific discussions on the evolution and change of AHC. As noted in the previous comment, the time periods are exactly the same as the ones selected for showing the exemplary trends in Fig. 4. We also note at this point that a typo in line 633 was corrected; the sentence is correctly saying now "highlighted in Figure 5 and summarized in Table 2" (rather than "in Figure 4").

- lines 656 and following and line 672 (still for AHC, settled on 1993-2020, 2001-2020 and 2006-2020?),

See previous two comments; these time periods for Table 2 are again exactly the same as used in Fig. 5 and introduced in Fig. 4 before.

- line 718 (now referencing ERA-Interim rather than ERA-5?),

This land-related analysis of Section 4 for estimation permafrost thawing, cited from Nitzbon et al. (2022), had used ERA-Interim reanalysis data for part of its surface climate forcing over the last four decades; a work independent from the AHC estimates reported in the Section 3. The sensitivity of the land heat storage estimation results to the specific choice of atmospheric reanalysis data is very small within the estimation uncertainties, though, since the heat increase from permafrost thawing itself is of a size of 2 ZJ, less than 10% of the total estimate dominated by continental heat storage (see discussion in last paragraph of Section 4).

- line 728 (Fig 6, land [continental?] Heat content 1960-2020,

Our goal was to present time series for all components for the period 1960 onwards if possible – which is limited for some (particularly for the cryosphere).
For the wording: we use: 'Heat to warm land', and we use 'continental heat strorage')

- line 790 (Fig 7, heat to melt ice, now using 1979 [first satellite measurements of sea ice?] to 2020),

Yes, as clarified in the text, this is due to measurement availability.

- at line 899 (specifying temporal coverage of satellite products 1980-2011 and 2011-2020), line 934 (total heat gain cryosphere now 1971-2020),

Yes, we are aligned with observing system availabilities.

- line 954 (total heat inventory 1971-2020 but summary Fig 8

Yes, we show what we could obtain from 1960 onwards, and as stated in the text we provide the trend values from 1971, i.e., aligned with IPCC periods.

- at line 964 clearly shows 1960-2020 and references to 1960),

We use the longest period possible for the reference period.

- line 1014 (now three periods 1985-2020, 1993-2020 and 2006-2020,

We start in 1985 only as we do not have TOA data available before. However, we have now decided to remove this figure to further avoid confusions. The results for the increase aspect of EEI will be discussed elsewhere (another publication in preparation).

- lines 1037, 1038 (now 2006-2020 represent a GCOS golden age where earlier we had those years as Argo "golden age",

Thank you, and we move this back to 'Argo golden age'

- lines 1089 and following (concluding at 1971-2020 and "recent era" 2006-2020),

This is the evaluation period chosen for the inventory.

- line 1130 (Fig 10 overview appropriately references in fig legend but now covering 1955-2020),

Not applicable anymore, figure removed.

- and line 1139 (excellent Fig 11 now back to 2006-2020 and 1971-2020).

Yes, core evaluation period.

Substantial effort to compile this list. We would not want to burden readers with same task, but they will need a clear summary! If composite inventory ends up covering 1971-2020 and 2006-2020, then same period should flow down to all individual components? But manuscript started with 1960-2020, 1993-2020 and 2006-2020? Authors need to provide guidance for readers to get from starting time periods to concluding time periods? Not clear at present! Given ocean dominance of global heat inventory, adopt ocean time periods throughout? However, recognizing different sources, reanalyses available for some components for some time periods, major role of satellite products for other components, etc., authors may not agree (or, have time to agree) on common time periods? If not, readers need a table or graphic somewhere to show major overlap periods. And, from abstract to conclusion, clear statement by authors of which time periods the inventory will cover and why. Even for a knowledgeable reader, this seems confusing at best. Help the readers with explicit clarification? Perhaps a time (multiple) line chart with observational emphasis periods for each component with final inventory periods super-imposed? Readers deserve some help, graphic or otherwise, to sort these multiple time periods.

Again, we would like to thank the reviewer for this support and comment. The strategy is to present for each component what is today available for each component, and to then move

ahead with one common period, which is additionally aligned with IPCC standards (e.g., 1971 onwards) on the one hand, and best available data on the other hand (from 2006 onwards). We have now established a clear statement at the bottom of the introduction, at the top of each section, and in the inventory section. Moreover, we have added a table for the cryosphere component for more clarity.

5) At many locations throughout text authors make reference to prior version; good! But (and thinking ahead) should authors in this version start a table to explicitly list and track changes/ improvements from prior version? Perhaps not so important for v2 but by v4 or v5 you would need such a list/table?

We thank the reviewer for this comment, and we agree that such a 'living table' for upcoming issues would be an assessed in an appendix. For this version we agree that more specific actions are needed, and hence we have added missing specifications in the final paragraph of the introduction.

**List of technical changes / questions:**

Line 80, abstract: Rewrite this after you made / considered all other changes, including shortening and removals? Needs work …. Present confusion about rates (W/m2/decade) vs imbalance amounts. Too much advocacy, particularly wrt GCOS?
Thank you, and we have revised the abstract, considering the reviewers comments, and comments from the second reviewer. We have improved the wording so that the use of the different rates is improved, and we have removed messaging about W/m2/decade, and we have removed GCOS advocacy, and hence link to the observing system in general.

Line 90: as noted, 2006-2020 does not constitute a decade. Check punctuation marks in 'et al.' Mostly correct but occasionally (e.g. line 96) not.
Thanks, we have changed decade to period, and checked the punctuation throughout the document.

Line 105, Introduction: Too much IPCC and GCOS text here. Start introduction instead at line 139 or even at line 163. Good place to save space by citing v1 as much as possible/ reasonable?
Thanks, and we have followed the recommendation, and strongly cutted – we have just kept 2 sentences from the 'GCOS paragraph' for context.

Line 167: Forster et al reference represents your primary much-used IPCC WG1 Energy Budget chapter; you know this but readers may not. On first use of Forster citation, label it as AR6 WG1 Energy Budget paper so that, in subsequent use of Forster et al. readers will (we hope) recognize it as the WG1 reference? E.g. here (or at line 142) use 'as itemized in energy chapter of AR6 WG1 (Forster et al., 2022)'. (Or something similar, authors will know best what they want.)
Thanks, but as we have now removed this sentence, and then the Foster ref. will appear within a list of others, it is challenging to follow the reviewers suggestion, and the link to IPCC is provided in the reference list.

Lines 189-192: Good statement, basic motivation of this product. This (or something similar) should appear in abstract?

We agree with the reviewer that this is a very important, and we have now included this statement in the abstract as well.

Line 195, Figure 1: Nice graphic but not needed, not useful, in this data description. Only four numbers here, without reference to time period. All numbers repeated multiple times in subsequent text, tables, figures. Save this one for poster or presentation? Not useful here.

We disagree with the reviewer, and do not fully follow the line of argument why this figure should be removed. Such a graphic has been developed as there had been a strong demand form a non-scientific audience, such as the science-policy interface, and makes nicely the case for the importance of the Earth heat inventory, and implications of heat accumulation in the Earth system. It provides the fully picture to introduce into the subject, which had been never published before, and we think that this figure will become an essential tool for this knowledge transfer. We hence would like to keep this graphic included. Also, it nicely underpins the multidisciplinary aspect of this inventory, and the multidisciplinary collaboration.

Line 214: Cheng et al. 2022. But, you have two Cheng et al. 2022 listed in reference, one in J. Climate and one under review at Nature. Not sure how Copernicus deals with papers in review but authors will need to specify via 2022a, 2022b?

Thank you, we have fixed that, and the paper is also now accepted.

Line 225: Permafrost warming / degradation does not only lead to ground subsidence. Might also lead to fracturing, ponding, heaves, erosion, etc. Better to say here: leads to disruptive changes in ground morphology?

The reviewer is right, permafrost thawing leads to a set of varied phenomena altering the land surface. We have included the suggested change in the new version of the manuscript.

Line 232: here you reference recent AR6 WG2 report (and cite as IPCC product). Something like this should also appear back at line 142 or 167? Need consistency in how you report AR6 chapters, e.g by author name or by IPCC chapter?

Thank you, and we follow the IPCC guide for citing the report, and to avoid further confusion, we have removed here 'as assessed in detail in the recent IPCC Working Group II report'.

Line 239 and following: This bulleted list is new to this version. Good summary, but high redundancy with preceding text? This reader finds the list useful, so perhaps scrub overall introduction to ensure minimal redundancy with this list.

Thank you for the comment. We agree, and hence we have now removed this list, and transferred it into an opening paragraph of the conclusion.

Lines 275 to 287: Good outline! Here, add explicit note (including, perhaps, a Table) of changes from prior version (see comment #5 above)? Note that outline includes discussion of future evolution of GCOS as a topic for the end conclusions / recommendations. Appropriate here (there) as opposed to too much discussion throughout introduction.

Thanks, and as already mentioned above, we have strengthened the information on 'what is new' here.

Line 291 and following: A lot of this text on history of ocean observations repeats almost verbatim what you wrote already in v1. Cite that version for all historical recounting; save this version for accurate detailed summary of current best data sources? Too much credit given here to GCOS, who basically passively benefited from Argos? Better to not engage in that discussion here?
We agree, and benefitted from cutting here.

Line 302: Introducing IQuOD here, new to this version (could have been listed in change table, comment #5) but not yet a community-wide standard. Hopeful, with notable goals, but not yet widely adopted? Only at V0.1? Many other versions of ocean thermodynamic data sets in play, e.g. GLODAP (much used and cited in ESSD); CLIVAR colleagues will know this field. Present status of IQuOD not clear to this reader, not needed here? Could list IQuOD as hopeful new QC product in conclusions/recommendations? Fits in category of promotion rather than of detailed data description? Same for ocean mass data: mention in Conclusions/Recommendations as future enhancement?
According to the recommendation above this has been removed here, and is more discussed in the provided references.

Line 315: Argo represents a remarkable community technology and data achievement. Credit GCOS for establishing ocean heat ECV and for advocating open access but, technically, GCOS did not initiate Argo system? E.g Argo system web pages may reference one or two ECV but carry few or no references to GCOS? Some overlap (post-project) in personnel but otherwise independent? Much of this discussion of ocean data developments could be cited from v1, rather than reproducing here?
This comment is well taken, and we have removed large parts of this paragraph here.

Line 372: Figure 2, excellent essential graphic, Uncertainty ranges, all nominally 2-sigma, appear much larger here than in v1? Due to different (more recent) climatology: 2005-2020 rather than 2005 to 2017 used previously? With different prior climatology period, would current anomaly uncertainties grow larger or smaller? No discussion or explanation provided? Note that here reader encounters three time periods with most recent (2006-2020) called "golden Argo era" (re: comment #4 above).
For version 1, 1-sigma had been used, whereas 2-sigma has been used for version 2. This is now coherently applied to all components, as well as the inventory.

Line 419: Trend analysis using LOWESS also new to this version (again, which Cheng et al. 2022?), this change could have been listed in change table?
We have precised this now in the final paragraph of the introduction.

Line 454, Table 1: Extra significant figure(s) in many numbers of this version compared to prior version? Not consistent with wider uncertainty ranges? No explanation? A consequence of LOWESS?
As explicitly explained in the text, we have moved to another method (LOWESS) and in addition, 2-sigma levels have been used.

Line 501, Section 3: Discussion of tropospheric thickening coupled with stratospheric cooling and shrinking represents another addition to this version. Good, but would also deserve mention in a 'changes' table?

Yes, we have updated the introductory discussion to the atmospheric section, to cite also newer relevant results that point specifics of atmospheric warming (we have now added one more very recent result again; Ladstädter et al. Sci Rep 2023).

And as mentioned above, we do not consider to include an overall change table in this version, as this appears to be fairly difficult to coordinate and coherently introduce to the present paper at this point now, but we definitely consider this for upcoming versions.

Line 525 and following: Much of this is identical to v1 but most readers will need to have these equations and this explanation at hand. No changes.

Ok !

Line 552: ECMWF-IFS, 2015 - this reference not defined!

The reference was inadvertently missing and is included in the list of references now.

Line 560: Additional changes described here that one could / should list in a 'changes' table? Changes include different treatment of JRA products, different use of radiosonde and radio-occultation products, drop the MSU, etc. All positive, but list them so readers will know! AHC figure (Figure 4) much improved in this version!

See answer to Line 501 comment above.

Line 670: Strongly agree with sentiment in this paragraph but I worry about the term "at an unprecedented rate". Unprecedented compared to what? To other components of the EEI? To undefined past time periods? Readers will need context here; I suspect you might need to change wording.

We agree that "unprecedented" should be used as the wording here only if there is a suitable (past-time context) reference. As such one is not yet available specifically for AHC (work including long-term natural variability estimations is prepared but not yet published), we toned down and changed to "has strongly increased over the recent decades."

Line 687 and many following lines in Section 4: Readers will appreciate and eventually read and use the many Cuesta-Valero references but - for the moment - most of those references remain submitted only. I suspect, as for other publishers, Copernicus does not allow 'submitted' references? Although many of the Cuesta-Valero references carry delineating characters (e.g. 2002a, 2020b, etc.) in the text, the same references remain very inconsistently referenced in the formal list. Need some serious fixes here.

Thanks for pointing this out, and this is fixed now. The references provided are papers which have been developed in parallel of this study, and to support the initiative but to assure at the same time visibility of those performing in-depth and new analysis, which should be published in an individual paper, and we are very grateful that these champion authors have undertaken this work at such a short time window. So we hope that the papers will become in an accepted from before publication. For format, we use the Mendelay tool, and further editing will be performed during the draft processing period.

Line 713: "generated using various global ground datasets" not detailed enough for a data journal? You want to provide information sufficient so that readers can duplicate your outcomes,
We have included the references for the datasets in the new version of the manuscript.

Line 718: the CSIRO land modeling effort relies on ERA-Interim where elsewhere (e.g. atmosphere) authors specifically chose ERA-5. Reconcile if possible?
We are aware that using ERA-Interim is a limitation of our estimate, as ERA5 is now available. Unfortunately, we cannot change the reanalysis version and redo all the simulations to estimate permafrost degradation now, but we plan to include this upgrade in the next iteration of the collaboration.

Line 750 and following: Confusing. Outcomes here use same input data as in v1 but produce the same patterns? In next sentence, similar outcomes derive from differences in processes? Awkward at best, most readers will need revision and clarification.
What we meant here is that the lower ground heat storage reported in this analysis is compensated by the newly added permafrost heat storage and inland waters heat storage, thus obtaining a value of total continental heat storage that is similar to the value in von Schuckmann et al. (2020). We have changed the text in the new version of the manuscript to improve the clarity of this part.

Line 767 and previous: Perhaps explain the term "continental" as it differs from 'terrestrial'? Readers may not understand the distinction and will not yet have access to Cuesta-Valero papers.
The paper explaining the estimates for continental heat storage is now available as a preprint in "Earth System Dynamics" (Cuesta-Valero et al., 2022b). In any case, we have included an explanation about the wording in the new version of the manuscript.

Line 782: Northern Hemisphere seasonal snow cover on land has a pretty good time series. Snow on ice (glaciers or sea ice) remains very difficult to quantify.
Thank you very much, and stated in the text we aim for inclusion for the next update.

Line 793: Reader encounters 95CI where earlier we saw 2-sigma. Similar uncertainty range but two different statistical naming conventions? Assumes normal distribution of random errors?
Thanks for spotting this, and we have adjusted at OHC level.

Line 802: As for prior equations, this equation should carry an ID number?
No numbering is provided for the formula as not further referenced to in the text.

Line 936: Strange punctuation around the von Schuckmann citation?
We apologize, but we do not understand the comment.

Line 955: obtained, obtained? Please revise.
Thanks, and fixed.

Lines 960, 961: "challenging to be quantified with respect to gaps in the observing system"? Not sure what the authors intend here?
Thanks for having spotted this – and we have revised.

Lines 975 to 977: clear statement here of EEI over 1971 to 2020 and 2006 to 2020. Please extend this clarity back to the abstract?
Thanks, and added.

Line 999: Awkward as written. Need caution - I agree. Perhaps: "Rate of heat accumulation across the Earth system'? Needs slight revision. Citation punctuation problems throughout the paragraph. Authors handled this point more accurately and more gracefully in v1.
Thanks, and we have followed the reviewer's suggestion. For the brackets, this is linked to challenges for Mendeley use, and we will exchange with the editor if this can be tackled for the final article processing.

Line 1019: very helpful figure with good explanation of time periods but we must regard 2006-2020 as the 'Argo golden era' as the author did in prior descriptions rather than a 'GCOS' golden age? GCOS itself general does not identify specific periods as 'golden ages'?
Thanks, and we have removed the wording 'GCOS'.

Line 1023: The opening phrase of this sentence, referencing GCOS, seems irrelevant to the remaining content of this paragraph.
Done and removed.

Line 1036 and following paragraph: Readers will not understand in this paragraph whether authors refer to generic global climate observations or to formal GCOS organization. If authors intend to add expressions of concern about GCOS with reinforcing statements about GCOS ECVs (for example) in their Conclusion / Recommendation section which follows (as this reviewer recommends) then this paragraph should refer more to the generic need for careful time series of precise observations with perhaps a final point about coordination by GCOS proper in the final sentence?
We have changed the wording from GCOS to observing system to avoid confusion.

Line 1059 and following: Provision of separate files for ocean, atmosphere, land and ice components seems useful and appropriate, with 5th file reporting the composite energy budget. This reader does not understand (and, does not remember reading a reason) why permafrost data exists in a separate file? Earlier we read about data held at DKRZ, presumably under DOI. Those long-term files will replace these short term Handle-labelled files? Someone, presumably at Copernicus, will ensure that transition? This reader notes that clicking on links results in error but that copy/paste of full Handle ID works. Registration barrier imposed by WDCC at DKRZ (similar to v1) approved by ESSD?
We have proposed the different author teams to publish their dataset individually if they wish to assure ore visibility to the working groups who have invested substantial amount of time in their contribution – which is not that clear for the long author list mostly in alphabetic order. So this way a chance had been provided to do so, and this is the why are some separated, and others not. In recognition of their support to this (non-funded), including early career scientists, international initiative we would like to keep this organized in this way. Concerning the IDs – yes, this is now fixed and replaced by the respective dois – an issue we faced to balance time schedules and data publication

delays. Thank you for having handled this situation for the review process, and we hope that with the dois the data publication part is more straightforward to access to.

Line 1084: Sentence beginning "Moreover, this study succeeded to improve" reads as awkward and run-on. Authors can do much better.
Thank you, and we have removed this part from the sentence.

Line 1100: Paragraph about Glasgow outcomes not relevant and not useful to most readers. Delete. Issue of including EEI in global stocktake addressed quite well in following paragraph.
We follow the reviewers advice and start the paragraph with the part on the paris agreement, and merged then with the following paragraph.

Line 1130: Figure 10, new to this version. Utility / information content not clear to this reader? If authors and/or editors intend to keep this graphic, it needs substantial revision (even for final downloaded version) to make it readable / legible. May remind many readers of a similar temporal evolution of sea level projections figure but without similar information impact. Combination of EEI total with OHC, while understandable, also adds confusion? Not a useful addition from the perspective of this reader.
We agree with the reviewer that this figure holds complexity. This from of synthesized assessment has received lot of positive feedback during the 2020 publication. We have however decided to remove the figure from this draft, and include it in an advanced version into another publication.

Line 1139: Figure 11 (formerly, Fig 8), excellent, comprehensive, a fine graphic take-home summary. Legend in prior version referenced needed CO2 reductions, not included here; I leave that one to authors.
Thank you. We have not included the link to the CO2 discussion in this version as we are working on another publication in parallel which is not yet ready to go, and hence we prefer to work on this study, and potentially include this discussion for v3.

The following paragraphs, outlining needs and recommendation for most (atmosphere, land, ice) component, seem helpful. This reader misses (and, authors miss a great opportunity to promote) an equivalent conclusion / recommendation focused on ocean? No shortage of ocean observation issues raised earlier; most of those would fit very well here! If authors wanted to delineate 'official' recommendations, those could start at line 1255.
Thank you, and we have made a focus for the ocean in the synthesis here.

---

## Author Comment (AC2)

**Point-by-point reply reviewer 2**

This study is an update of von Schuckmann et al. (2020) heat inventory. It provides 2 more years of the inventory from 2018 to 2020. There is one innovation compared to von Schuckmann et al. (2020): the new heat inventory includes now estimates of the permafrost thawing, inland freshwater and Antarctic sea ice heat uptake. In this paper, the authors call for a regular update of their heat inventory and for an implementation of the heat inventory in the Paris agreement's global stock take.

This manuscript is dealing with a very important aspect of climate change: the heat uptake of the climate system. The paper is well written and easy to follow. The methods used are sound.

Scientifically speaking, I am disappointed by this manuscript. I find the progress compared to von Schuckmann et al. (2020) is incremental and the results are not new. The uncertainties are not improved compared to von Schuckmann et al. 2020 (not better documented and not reduced in any manner either) and we don't get substantial new knowledge out of the analysis that are proposed.

However, in terms of climate policy and knowledge for action, I think this paper is relevant and support an important position in the community. I definitely agree with the authors that the heat inventory should be implemented in the Paris agreement's global stock take and should be more advertised to the general public. I find this manuscript supports nicely and efficiently this position.

In summary, I find that this paper is more a position paper than a scientific paper. I think the authors should acknowledge that and be clearer on this aspect. I also think the authors should target journals that are more suitable for position papers. By publishing in ESSD they may miss a substantial part of their targeted audience.

We have chosen ESSD because it allows for concurrent data publication, open review process, and recognition for similar regular reporting approaches such as the global carbon project. We hence believe that this journal is a choice taken to balance between science needs and transfer to a wider audience, and allow for the concurrent publication of the underlying data set.

We thank the reviewer for the comment, and the overall review, and we hope that the revised version of the paper will meet expectations. We would like to stress that this draft is intended to provide an update of the previous pilot study. A point-by-point reply is provided below.

**Detailed comments:**

L139-143: I find this picture of the heat accumulated in the Earth system, which would result from anthropogenic GHG emissions only, too simple and misleading. I think you should acknowledge there is a more complex situation here. At least you should mention the role of other important forcing such as the aerosol forcing and the role of internal variability as well.

We agree with the reviewer about needed revisions for this part of the introduction, and together in reply to comments of reviewer1 and this reviewer, we have now proposed a major revision for this part, and the second paragraph now directly goes into the complexity explanation as mentioned by the reviewer.

L151: you probably mean "confirmed" rather than "revealed". The long-term heat gain has been revealed a long time ago (ex. Levitus et al. 2001)

We agree, but not relevant anymore as text had been now removed.

L160-161 : indeed the results are closely consistent with the IPCC AR6 and von Shuckmann et al. 2020. I don't see here any significant improvement compared with previous estimates. The improvement only comes from the addition of two more years but the picture of the heat redistribution has not changed. I find this improvement is really incremental compared to von Shuckmann et al. 2020

We thank the reviewer for leveraging the major challenge of regular updates for a climate change indicator which will not reveal fundamental new advancements in science, but rather complements with each update the full picture of the current capacity of estimate, remaining challenges, and the current state and quantification of the EEI, and the Earth heat inventory. Moreover, this initiative allows for international collaboration, bringing together experts across all fields of climate research, and raises new discussions and research questions. As stated by the reviewer above, this update has succeeded to increase the collaboration for the cryosphere component, and to connect to communities for permafrost, and inland freshwater. New publications have been submitted in parallel to this work, and new research discussions are under the way. As for the global carbon project, we believe that this community momentum is of great value for the climate research community.

L185: To my knowledge ice shelf mass discharge has never been attributed to anthropogenic GHG emissions so far (although the attribution is highly probable). This is because attribution needs a thorough understanding and modelling of the processes at play which is not yet available for ice shelf. So I suggest to remove "ice shelf" from this sentence.

Thank for your comment. We have followed the information from IPCC AR6 for the concept of committed change, for which ice shelf counts to, see Foster et al., 2021.

L258 : I don't understand why the heat inventory provides a tool for assessing the general status of the GCOS. Can you elaborate ?

Due to recommendations of reviewer 1, this part has been removed from the introduction.

L266 : Any other climate indicator or scientific study enables « concerted international and multidisciplinary collaboration ». I don't see a special added value from the heat inventory over other initiatives.

Obsolete as text has been removed.

L269-273: I think these lines are the core of this paper. I understand you are calling for a regular monitoring of the heat inventory to support the IPCC solution pathways and to support regular stock taking of the implementation of the Paris Agreement. So this paper is more a position paper than a scientific paper reporting on recent progress. I think this aspect should be assumed from the beginning and the paper should be presented as a position paper rather than a scientific contribution.

We hope that with the proposed revisions we could follow the advice.

L310-313 : OHC estimates from remote sensing through the global sea-level budget are not merely "possible". They are now mature (See Hakuba et al. 2021, Marti et al. 2022). You should consider these estimates here.

Thank for this comment, and we agree that the current formulation is mis-leading, and further information are missing. We have interacted with this group of experts, and now experts are onboard as co-authors. Accordingly, and after advice from the additional experts, we have now added more text, and discussion. In addition, we have now included the satellite full-depth estimate for the most recent period (2006-2020 and 2006-2019), and compare it to the in situ full-depth estimate (see table 1, and text).

L352: the problem in using the spread as a proxy for uncertainty is that you don't know the sources of uncertainty. Can you tell us more about the uncertainty here? What are the main sources of uncertainty? Which one dominates? What is the temporal structure of the uncertainty? Is it correlated in time? Do you consider this information to compute the tendency?

Thank you for the comment. We have discussed these different aspects at different places in this section. We have now better grouped the information, including the knowledge obtained on the different sources of uncertainty according to previous studies, and better clarified the fact that the different mapping approaches and the choice of the climatology are a major player, together with the bias correction approaches for the historical time series. Relevant references are provided. For the trend estimate, a large number of sensitivity tests have been performed on advance as part of a Ph.D. thesis which are about to be published elsewhere (still draft development stage). Results from this study indicate in agreement to the study of Cheng et al., 2022 that for this approach, the use of LOWESS as discussed in Cheng et al., 2022 has been used, together with a monte-carlo approach for the uncertainty range. This is well discussed in the text. We have additionally also highlighted now better the important use of the so called 'synthetic profiles approach (Allison et al., 2019), and future evaluations are needed for a more in-depth evaluation of the uncertainties which is however out of the scope for this study.

L402: what does "largely homogeneous criteria" mean? Please be specific

Thanks, yes, we agree that this sounds awkward, and we have removed 'largely', and added a double point at the end of the sentence to indicate that the criteria follow.

L416 : do you mean « of the corresponding ensemble"?

Yes, and changed accordingly.

L430: time correlation in the uncertainty could bias significantly your trend estimate. Have you considered this?

Thank you for this question, and no, with the ensemble approach we are not able to consider this aspect, and this would need to undergo a systematic study (e.g., based in the systematic profile approach, e.g., Allison et al., 2019) such as discussed in the paragraph above, and according to the reply to the reviewer's question above.

L448: by "below 700m" you mean between 700m and 2000m depth, right? Please be specific

Thanks, and yes, we agree, and it has been changed accordingly, now clearly referring to the 700-2000m depth layer.

L529: there are typos in the equation. Please correct it

Thank you, corrected, and also one variable naming improved plus related small text edits implemented around (the import of this equation into the joint manuscript inadvertently had led to partial loss of characters).

L618 Figure 4: same remark as before. You are using the spread as a proxy for uncertainty. The problem is that you do not know the sources of uncertainty. Can you tell us more about the uncertainty here? What are the main sources of uncertainty? Which one dominates? What is the temporal structure of the uncertainty?  Is it correlated in time? Do you consider this information to compute the tendency? What about systematic sources of uncertainty?

Yes, we use the spread as reasonable proxy for the overall uncertainty captured by these multiple atmospheric datasets and discuss key aspects of uncertainty sources, respectively the long-term quality, in Subsection 3.2 related to the input datasets. Here we particularly refer to key references both for the reanalysis and observational datasets, where the dataset providers have characterized individual dataset quality in more depth as well as discuss why we left out some older "outdated" datasets, which would unduly increase spread given they are known from cited sources to be of inferior quality. In the last paragraph of Subsection 3.2 we also discuss that the differences of sampling between observational and reanalysis datasets are a minor source of uncertainty in the resulting AHC estimates (i.e., much smaller than the ensemble spread). The discussion of key aspects how the uncertainties that make up the spread in AHC gains (i.e., of the trend fits to the AHC anomaly data) is part of Subsection 3.3, where especially the fairly large spread induced in latent AHC gain and its uncertainty sources is discussed, mainly rooting in observational (RS, RO) differences as discussed and, in general, also in natural variability.
To further improve the discussion, we have now included an additional last paragraph in Subsection 3.2 to summarize for these atmospheric datasets the role of the ensemble spread as uncertainty measure, which is in fact of minor relevance for the AHC trend (i.e., AHC gain) uncertainties which are dominated by interannual natural variability, plus special characteristics of time-dependent systematic effects in individual datasets as discussed in Subsection 3.3 (e.g., discussed for the RS data; time-constant systematic errors, or biases, would anyway play no role in these anomaly time series analyses that focus on time changes and temporal trends).

L667-670 I agree with the authors, the study here essentially confirms von Schuckmann et al. 2020

OK.

L759-760: I understand the much lower uncertainty in ground heat uptake in this study is coming from the new inversion method for the vertical temperature profiles. Why should we trust this new uncertainty estimate rather than von Schuckmann et al. 2020? What makes it superior?

There is a publication now, Cuesta-Valero et al. (2022a), explaining all details of the new inversion method, including a thorough comparison with the inversion method used in von Schuckmann et al. (2020), and demonstrating that the previous inversion method leads to the new uncertainty results when standard error propagation is applied. Cuesta-Valero et al. (2022a) shows that the previous technique used to aggregate inversions from individual profiles was markedly conservative, overestimating the 95 % confidence interval for the global mean inversion. Therefore, we consider the uncertainty reported here to be more robust than the uncertainty estimated in von Schuckmann et al. (2020).

L929: Why attributing the same uncertainty to GIOMAS as to PIOMAS? Is it reasonable? Why so?

Thank you for the comment, and according to the reviewer's comment we have now revised the draft to *'In the absence of a detailed characterization of uncertainties for these estimates, we use the uncertainty in GIOMAS sea-ice thickness of 0.34 m (Liao et al., 2022) to estimate the uncertainty in GIOMAS sea-ice volume to be ±4.0x10^3 km^3, fo which have used an annual mean sea-ice extent of 11.9x10^6 km^2 (Lavergne et al., 2019). One caveat to this is that the observational estimates have their own significant uncertainties (Kern et al., 2019; Liao et al., 2022)."*

L 1032: how do you estimate the rate in EEI and the associated uncertainty . I don't understand how you can get such a small uncertainty in the rate of change of EEI when you have such large uncertainties in the estimate of heat uptake of different components of the Earth system. Please, detail your uncertainty estimate here?

This part, together with Fig. 9 have been now removed according to the review process.

L1044: I disagree. Fig 9 shows that the primary need is to reduce uncertainties rather than to extend the time series.  Can you comment on this? Why do you put forward the extension while uncertainties are still so large?

This part, together with Fig. 9 have been now removed according to the review process.

L1088: not "reveal" rather "confirm"

Thank you for this comment, but we do not agree. This is a new result according to this analysis approach, and for a different period, with hence continued heat accumulation. TO our knowledge, no heat inventory is today published up to 2020.

L1100-1127: I agree with this paragraph and I agree this is important to call for implementation of the Earth heat inventory into the global stock take. But I don't think it should be done in a scientific technical paper. It should rather be done in a scientific position paper. In addition, ESSD is probably not the best place to do that.

We would like to thank you. The journal webpage states: 'Earth System Science Data (ESSD) is an international, interdisciplinary journal for the publication of articles on original research data (sets), furthering the reuse of high-quality data of benefit to Earth system sciences. The editors encourage submissions on original data or data collections which are of sufficient quality and have potential to contribute to these aims. '

We believe that this paper provides a rationale for the datasets we publish with this article, driven and described by the international community on the Earth heat inventory and their different components. And we are convinced that these research data sets are of benefit to Earth system sciences.

L1147-1267: How come there are no recommendations on improving/reducing uncertainties? Figure 9 is probably the most advanced scientific result of this paper and it definitely calls for a reduction of uncertainties. I suggest to put some recommendation along these lines at a high level of priority. If not, we would like to understand why uncertainties are ignored

Thank you for the comment, and we fully agree with the reviewer, and have now added several sentences in the conclusion accordingly.

L1269: Here again, I find the position of the paper is not clear. If this is a scientific paper dedicated to scientists (as ESSD is for) then updating the record is not so important. Reducing uncertainties is probably much more of a priority. But, if this paper is a position paper more oriented toward climate services which calls for the implementation of the heat inventory in the global stock take then yes the priority is probably to update regularly the record. In the latter case the manuscript is probably proposed to the wrong journal and I am afraid you may miss your targeted audience.

As replied in the comment further above, we think that the inclusion of this update is justified for ESSD, and such type of regular updates are provided also for other indicators (e.g., the global carbon budget). This paper is not only a perspective piece, but it brings together international and multidisciplinary expertise on the different Earth system components. Also, new estimates from the science community have been added for this second update, and there is today no information in science literature available on the different Earth system components up to 2020. This evaluation cannot be picked up currently from a climate service for example for regular operational update as scientific expertise and analysis is needed to provide the quantifications based on joint data and model studies, with for example 2 scientific papers currently under review have been needed to provide a contribution for some estimates. Moreover, these time series for all Earth system components have been made available for the science community (and all), and we are convinced that they will become of important value for new scientific studies, and climate model evaluation purposes. Finally, our discussions and results also provide a fundamental foundation for observing system recommendations. These results lead into a science-driven provision of climate data relevant for climate research, and for the use of climate change reporting.

---

## Author Response (AR2)

**Public justification (visible to the public if the article is accepted and published)**:
Essential compilation; thanks for using ESSD.

A few suggestions, yours to accommodate or not, in this version or next. Cumulative changes should still result in fewer total lines/words, always a positive for ESSD:
*We would like to thank you for the opportunity of further adjustments for the manuscript thanks to these comments, and we provide a point-by-point reply below.*

Lines 183, 184: EEI causes increase in Global Mean Surface Temperature (GMST). New definition? Not used again in this manuscript? Cited elsewhere? Later (line 189) you mention heat accumulation warming the (full tropospheric) atmosphere. Redundancy?
*Thank you for the comment, and we have removed the abbreviation as it is not used again in the manuscript. For the second part of the comment we did not add further change as the GMST part is linked to the surface response, and the L189 aspect is linked to the impact of heat accumulation on atmospheric warming.*

Line 197: decrease in 'lake' ice cover? At line 657 you call them 'inland' water bodies.
*Thank you, and we have included 'lake'.*

Line 200 (and many following examples): please check punctuation as introduced by bibliometrics software. The example here seems wrong.
*We have used Mendeley for the references, and we hope that there is further iteration possible during the editing process.*

Line 202, 203: here you summarize changes in terrestrial, freshwater and ocean ecosystems but you have already listed some of these changes (stratification, algal blooms, microbial emissions from permafrost, etc. Do not repeat unless in clarification or by way of summary?
*We have removed 'terrestrial, freshwater and ocean' to remove repetitions.*

Line 222: We present, rather than 'we will present'? Affirmative!
*Yes, agreed, and changed accordingly, thanks for having spotted this!*

Line 241: Do you really mean past century sensu stricto? E.g back to 1920? Or during current century, e.g. back to 2000?
*Agreed, and we have changed to 'decades'.*

Line 247: Small changes (reduced text) can improve fit between statement and references. E.g end sentence after 'methodologies' then move to list of references.
*Thank you, and done.*

Line 257: Second time to mention global Argo by 2006 in one paragraph. Reduce?
*Thanks and we have removed '2006'.*

Line 444: Un-needed extra words here. Use instead 'small compared to other Earth subsystems' Too much redundancy in this paragraph; please reduce. With careful re-writing, this entire introduction to atmospheric heating could reduce by 20%?
*We have reduced the first paragraph according to the comment, but we think that the following parts are essential for further discussion in this section, and not yet discussed in the introduction.*

Line 501: most of us consider reanalyses as a necessary model assimilator of observational data, with periodic outputs (e.g ERA5) as QC-d observations in gridded format (you specify this starting from line 506). If so, this apparent distinction between 'observation-based and reanalyses' seems confusing?
*Thank you and we have removed this accordingly.*

Line 560: sorry to toot my own horn, but original identification (jointly with Vaisala) of humidity bias in RS80 comes from Wang et al. 2002? Holger will admit that his work (cited here as Vömel et al., 2007) derives from earlier work. (Too) Many people on 2002 paper no longer active (or, alive) but we should ensure they (especially June) get proper credit? Lines 619, 620 repeat the caution about RH bias? You do not need both?
*Thank you for pointing to this reference. We included Wang et al. 2002 in the reference list and cite it in the text. On the caution about the RH bias, we would like to keep it because we find it important to state it at this place again.*

Line 745: here again authors refer to 'lake ice cover' which they have earlier itemized as 'inland' ice cover. Please clarify, using consistent terminology.
*Thank you and we have replaced lake and river ice by inland water body ice (e.g., lake, river).*

For this reader, a collection of system-specific recommendations, now found scattered among earlier sections, would very much complement the more generic recommendations around line 1220. You would not need to rewrite, merely copy from source and paste near here (near Figure 9) somewhere. Not decorative, but serious: what you recommend or need to close important uncertainty gaps. A list to check against in future versions?
*Thank you for this comment and according to a reviewer comment, and this comment we have already started a detailed table internally to track the specific updates from the 2020 paper compared to this study which we aim to include in an appendix-style for future versions.*